

# Data assimilation for moving mesh methods with an application to ice sheet modelling

Bertrand Bonan[1], Nancy K. Nichols[1], Michael J. Baines[1], and Dale Partridge[1]

[1]School of Mathematical, Physical and Computational Sciences, University of Reading, Reading, United Kingdom

*Correspondence to:* Bertrand Bonan (b.c.bonan@reading.ac.uk)

**Abstract.** We develop data assimilation techniques for nonlinear dynamical systems modelled by moving mesh methods. Such techniques are valuable for explicitly tracking interfaces and boundaries in evolving systems. The unique aspect of these assimilation techniques is that both the states of the system and the positions of the mesh points are updated simultaneously using physical observations. Covariances between states and mesh points are generated either by a correlation structure function in a variational context or by ensemble methods. The application of the techniques is demonstrated on a one-dimensional model of a grounded shallow ice sheet. It is shown, using observations of surface elevation and/or surface ice velocities, that the techniques predict the evolution of the ice sheet margin and the ice thickness accurately and efficiently. This approach also allows the straightforward assimilation of observations of the position of the ice sheet margin.

## 1 Introduction

From lava flows to tumour growth, including water floods, many time-evolving processes can be mathematically modelled by moving boundary problems. Predicting their evolution accurately requires not only the estimation of the state variables of the system over a moving domain, but also the estimation of the location of the moving domain itself. In this paper we propose to combine data assimilation with a moving mesh numerical model to estimate both the domain and the states of moving boundary problems. The application of the combination is demonstrated on a one-dimensional model of a grounded shallow ice sheet. Our approach is particularly relevant to the prediction of the dynamics of ice sheets and glaciers. Future evolution of ice sheet boundaries is closely linked with sea level rise (Church et al., 2013) and ice sheets are now relatively well observed bodies (Vaughan et al., 2013).

Data assimilation (or DA) aims to combine available observations with model predictions in order to provide optimal estimates of the state of a system and an estimation of the uncertainty of these estimates. DA has been applied succesfully in various contexts and is routinely used in operational systems such as numerical weather prediction systems (Lahoz et al., 2010; Blayo et al., 2014). In particular DA has already been used with fixed-grid models in the context of moving boundary problems. In these cases estimates outside the moving domain are generally non-physical and need to be reanalysed. For example, negative sea-ice concentration and thickness can be obtained where there is no sea ice (Mathiot et al., 2012). The same problem appears with negative estimated thickness of ice sheets (Bonan et al., 2014). In both cases non-physical negative variables are reset to zero after data assimilation. Furthermore, with fixed grids, DA does not provide any explicit estimate of the extent of





the domain. This can be only done by interpolation. By combining DA with a moving mesh numerical model, this paper shows that the extent of the domain can be estimated explicitly and non-physical estimates do not appear.

Genuine moving mesh methods use a fixed number of mesh points whose movement can be generated by various techniques (Budd et al., 2009; Baines et al., 2011). In this paper, we model the evolution of a radially-symmetrical continental ice sheet

with a moving mesh-point method based on conservation of local mass fractions (Baines et al., 2005, 2011; Partridge, 2013; Lee et al., 2015; Sarahs, 2016). Such a method has no requirement in terms of initial distribution for mesh points and has been applied in various contexts (for example chemical spreading, Lukyanov et al., 2012, and tumour growth, Lee et al., 2013).

This paper proposes to adapt two popular DA schemes: a 3D-variational scheme (or 3D-Var, see e.g. Lorenc, 1986; Nichols, 2010) and an Ensemble Transform Kalman Filter (or ETKF, see Bishop et al., 2001; Hunt et al., 2007) to the estimation of the

state of an ice sheet modelled using the moving point method detailed in Bonan et al. (2016). The approach is validated by twin experiments using available classical surface observations (surface elevation and surface velocity, see Vaughan et al., 2013). Observations of the position of the moving boundary (see e.g. Dyke and Prest, 1987 for observations of continental margins in palaeoglaciology) are also assimilated by using a straightforward observation operator. The paper is organised as follows: in Sect. 2 we recall the key points of the moving point ice sheet model, in Sect. 3 we describe how to use the 3D-Var and

the ETKF for our state estimation problem, in Sect. 4 and 5 we validate our approach by performing several twin experiments before concluding in Sect. 6.

## 2 Moving-point ice sheet model

### 2.1 Ice sheet dynamics

We consider a single phase, radially-symmetric, grounded ice sheet (no floating ice), centred on the origin $r = 0$ of the radial

coordinates. The origin is called the ice divide.

The geometry of the grounded ice sheet is described by its surface altitude, $s(t, r)$, the ice thickness, $h(t, r)$ and the altitude, $b(r)$, of the fixed bedrock on which the ice sheet lies (see Figure 1). These quantities are linked through the relation

$$s = b + h.  \tag{1}$$

The position of the edge of the ice sheet $r_l(t)$, also known as the ice sheet margin, is implicitly determined by the Dirichlet

boundary condition

$$h(t, r_l(t)) = 0.  \tag{2}$$

The evolution of an ice sheet is governed by the balance between the mass exchanges at the surface (snow precipitation and surface melting) and the ice flow that carries the ice from the interior of the ice sheet towards its margins. This is summarised by the mass balance equation

$$\frac{\partial h}{\partial t} = m(t, r) - \frac{1}{r} \frac{\partial (r h U)}{\partial r},  \tag{3}$$



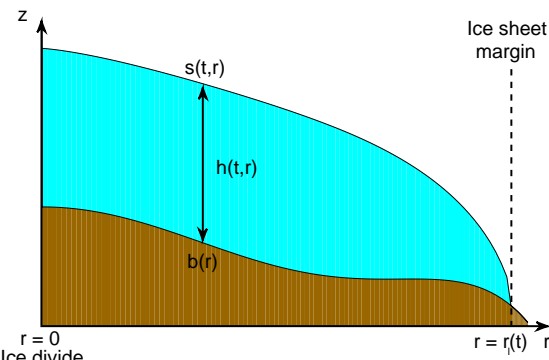

**Figure 1.** Section of a grounded radially-symmetrical ice sheet.

where $m(t,r)$ is the surface mass balance and $U(t,r)$ is the vertically averaged horizontal component of the ice velocity in the sheet. In the numerical experiments (see Sect. 4 and 5) we use two different surface mass balances: a function that only depends on the radius $r$ and a more complex surface mass balance which depends on the atmospheric temperature that evolves with the geometry of the ice sheet. Both surface mass balances are described in detail in Appendix A.

5     The velocity of the ice is derived using the Shallow Ice Approximation (Hutter, 1983), which leads to the following analytical formulation of the vertically averaged horizontal component of the ice velocity $U(t,r)$:

$$U = -\frac{2}{n+2}A\left(\rho_i\,g\right)^n h^{n+1}\left|\frac{\partial s}{\partial r}\right|^{n-1}\frac{\partial s}{\partial r},\tag{4}$$

where $s$ is given by Eq. (1) and the parameters involved in the Shallow Ice Approximation (SIA) are summarised in Table 1. Since we consider only radially symmetrical ice sheets, a symmetry condition also holds at $r = 0$

10   $U(t,0) = 0 \qquad \text{and} \qquad \dfrac{\partial s}{\partial r}(t,0) = 0.$          (5)

**Table 1.** Parameters involved in the computation of the vertically averaged horizontal component of the ice velocity (Eq. 4).

| Parameter | | Value |
|---|---|---|
| $n$ | exponent of the creep relation | 3 |
| $A$ | coefficient of the creep relation | $10^{-16}$ Pa$^{-3}$.yr$^{-1}$ |
| $\rho_i$ | density of ice | 910 kg.m$^{-3}$ |
| $g$ | gravitational acceleration | 9.81 m.s$^{-2}$ |

## 2.2   Moving-point method

The moving-point numerical method we use in this paper relies on the computation of point velocities and point locations. This type of method belongs to the family of velocity-based (or Lagrangian) methods (Cao et al., 2003). Here the velocity of mesh



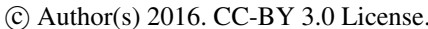



points is obtained by conserving local mass fractions (Baines et al., 2005, 2011). To calculate the velocity we first define the total volume of the ice sheet $\theta(t)$ as

$$\theta(t) = 2\pi \int\limits_{0}^{r_l(t)} r\, h(t,r)\, dr\,. \tag{6}$$

Assuming that the flux of ice through the ice sheet margin is zero, its rate of change $\dot{\theta}$ depends only on the surface mass balance,

$$\dot{\theta}(t) = 2\pi \int\limits_{0}^{r_l(t)} r\, m(t,r)\, dr\,. \tag{7}$$

We now define the relative mass fraction $\mu(\hat{r})$ relative to the moving point $\hat{r}(t)$. Since the density of ice $\rho_i$ is assumed constant, volume fractions and mass fractions are equivalent and

$$\mu(\hat{r}) = \frac{2\pi}{\theta(t)} \int\limits_{0}^{\hat{r}(t)} r\, h(t,r)\, dr\,. \tag{8}$$

The velocity of the moving point $\hat{r}(t)$ is defined implicitly by keeping $\mu(\hat{r})$ constant in time. By differentiating Eq. (8) with respect to time using Leibniz' integral rule, we obtain the velocity of every interior point

$$\frac{d\hat{r}}{dt} = U(t,\hat{r}(t)) + \frac{1}{\hat{r}(t)h(t,\hat{r}(t))} \left( \mu(\hat{r}) \int\limits_{0}^{r_l(t)} r\, m(t,r)\, dr - \int\limits_{0}^{\hat{r}(t)} r\, m(t,r)\, dr \right)\,. \tag{9}$$

One of the points is dedicated to the static ice divide $r = 0$, while another point tracks the position of the margin $r_l(t)$, which moves at the velocity (Bonan et al., 2016)

$$\frac{dr_l}{dt} = U(t,r_l(t)) - m(t,r_l(t)) \left( \frac{\partial h}{\partial r} \right)^{-1}\,. \tag{10}$$

Once the velocity of each moving point has been obtained from Eq. (9) or Eq. (10), the moving points are moved in a Lagrangian manner using the explicit Euler scheme

$$\hat{r}(t + \Delta t) = \hat{r}(t) + \Delta t \frac{d\hat{r}}{dt}\,. \tag{11}$$

The total mass $\theta(t)$ is updated in the same way using $\dot{\theta}(t)$ from Eq. (7). Finally, the ice thickness profile is updated by

differentiating Eq. (8) with respect to $\hat{r}$ giving

$$h(t,\hat{r}(t)) = \frac{\theta(t)}{\pi} \frac{d\mu(\hat{r})}{d(\hat{r}^2)}\,. \tag{12}$$

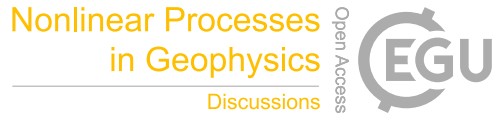

## 2.3 Numerical model

From the equations detailed in Sect. 2.2, a finite difference algorithm is derived (see Bonan et al., 2016, for the full algorithm). The mesh consists of $n_r$ moving nodes with the positions

$$0 = \hat{r}_1 < \hat{r}_2 < \ldots < \hat{r}_{n_r-1} < \hat{r}_{n_r} = r_l(t). \tag{13}$$

No further assumption is made on the spatial distribution of the moving nodes. At each node $\hat{r}_i$ there is an associated ice thickness $h_i$ and a fixed mass fraction $\mu_i$. By construction, $\mu_1 = 0$, $\mu_{n_r} = 1$ and the ice thickness at the ice sheet margin $h_{n_r} = 0$.

The user provides the initial mesh and the ice thickness at mesh points in order to initialise the numerical model. From these quantities, the initial total mass and the mass fractions are calculated by discretising Eq. (6) and Eq. (8) using the following composite trapezoidal rule

$$\theta = \frac{\pi}{2} \sum_{i=1}^{n_r-1} (h_i + h_{i+1})(\hat{r}_{i+1}^2 - \hat{r}_i^2), \tag{14}$$

$$\mu_1 = 0, \quad \mu_{i+1} = \mu_i + \frac{\pi}{2\theta}(h_i + h_{i+1})(\hat{r}_{i+1}^2 - \hat{r}_i^2), \, i = 1, \ldots, n_r - 1. \tag{15}$$

## 3 State estimation of a system modelled with a moving mesh

We now recall the basics of data assimilation before explaining how to adapt the 3D-Var and the ETKF methods to our context. We then clarify the form of the observation operator for various types of observations that we assimilate.

### 3.1 Data assimilation

We consider data assimilation in a discrete dynamical system evolving in time. We denote by $\mathbf{x}_k$ the vector of size $n_{\mathbf{x}}$ describing the state of the system at time $t_k$. For example, in our numerical ice sheet model, ice thickness at mesh points are elements of the state vector. The state $\mathbf{x}_k$ is propagated forward in time to a time $t_{k+1}$ by the nonlinear model $\mathcal{M}_{k,k+1}$. Assuming the model is perfect, we have

$$\mathbf{x}_{k+1} = \mathcal{M}_{k,k+1}(\mathbf{x}_k). \tag{16}$$

Observations are available at times $t_k$ and are related to $\mathbf{x}_k$ through the equation

$$\mathbf{y}_k = \mathcal{H}_k(\mathbf{x}_k) + \varepsilon_k, \tag{17}$$

where $\mathbf{y}_k$ is a vector of $p_k$ observations taken at time $t_k$, $\mathcal{H}_k$ is the, possibly nonlinear, observation operator and $\varepsilon_k$ is the observation error vector, which is assumed to be unbiased (zero mean) with covariance matrix $\mathbf{R}_k$.

The objective of DA is to provide an optimal estimate $\mathbf{x}_k^a$ of the system, called the *analysis*, by combining observations with information derived from the model. We consider in this paper two different DA schemes: a 3D-Var scheme and an ETKF.



### 3.1.1 3D-Var

3D-Var (see e.g. Lorenc, 1986; Nichols, 2010) aims to provide the optimal estimate $\mathbf{x}_k^a$ by minimising the cost function

$$\mathcal{J}(\mathbf{x}) = \frac{1}{2} \left( \mathbf{x} - \mathbf{x}_k^b \right)^T \mathbf{B}_k^{-1} \left( \mathbf{x} - \mathbf{x}_k^b \right) + \frac{1}{2} \left( \mathbf{y}_k - \mathcal{H}_k(\mathbf{x}) \right)^T \mathbf{R}_k^{-1} \left( \mathbf{y}_k - \mathcal{H}_k(\mathbf{x}) \right), \tag{18}$$

where $\mathbf{x}_k^b$ is a *prior*, or background, estimate of the state of the system (generally obtained by propagating forward in time the

previous analysis $\mathbf{x}_{k-1}^a$ with Eq. (16)). The error in the prior estimate is assumed to be unbiased with covariance matrix $\mathbf{B}_k$.

We take the observation operator $\mathcal{H}_k$ to be linear around $\mathbf{x}_k^b$, meaning that

$$\mathcal{H}_k(\mathbf{x}) \approx \mathbf{x}_k^b + \mathbf{H}_k \left( \mathbf{x} - \mathbf{x}_k^b \right), \tag{19}$$

where $\mathbf{H}_k$ is the linearisation of the observation operator about the background $\mathbf{x}_k^b$. Under this assumption, the cost function
has an explicit minimum

$$\mathbf{x}_k^a = \mathbf{x}_k^b + \mathbf{K}_k \left( \mathbf{y}_k - \mathcal{H}_k \left( \mathbf{x}_k^b \right) \right), \tag{20}$$

where

$$\mathbf{K}_k = \mathbf{B}_k \mathbf{H}_k \left( \mathbf{H}_k \mathbf{B}_k \mathbf{H}_k^T + \mathbf{R}_k \right)^{-1}. \tag{21}$$

In theory the true background error covariance matrix $\mathbf{B}_k$ should be updated at each time step. However, this process is
extremely expensive for real-time applications and, instead, we use a matrix with a simplified structure specified by the user.

We will see in the numerical experiments (Sect. 4 and 5) how setting $\mathbf{B}_k$ appropriately is essential in order to obtain good
estimates.

### 3.1.2 Ensemble Transform Kalman Filter

The Ensemble Kalman Filter (EnKF) introduced by Evensen (1994) aims to approximate the Extended Kalman Filter us-
ing a Monte Carlo method. At each time step, the state of the system is represented by an ensemble of $N_e$ realisations

$\left\{ \mathbf{x}_k^{(i)}, i = 1, \ldots, N_e \right\}$. The state estimate is given by the ensemble mean

$$\overline{\mathbf{x}}_k = \frac{1}{N_e} \sum_{i=1}^{N_e} \mathbf{x}_k^{(i)} \tag{22}$$

and the state error covariance matrix by the ensemble covariance matrix

$$\mathbf{P}_{e,k} = \frac{1}{N_e - 1} \mathbf{X}_k \mathbf{X}_k^T \tag{23}$$

where $\mathbf{X}_k$ is the anomalies matrix defined as

$$\mathbf{X}_k = \left[ \mathbf{x}_k^{(1)} - \overline{\mathbf{x}}_k, \ldots, \mathbf{x}_k^{(N_e)} - \overline{\mathbf{x}}_k \right]. \tag{24}$$



From the ensemble covariance matrix we can define the matrix $\mathbf{Corr}$ that contains an estimate of the correlation between the state variables to be

$$[\mathbf{Corr}]_{i,j} = \frac{[\mathbf{P}_{e,k}]_{i,j}}{\sqrt{[\mathbf{P}_{e,k}]_{i,i}\,[\mathbf{P}_{e,k}]_{j,j}}}\,, \tag{25}$$

where $[\mathbf{Corr}]_{i,j}$ and $[\mathbf{P}_{e,k}]_{i,j}$ denote the entry in the $i$-th row and $j$-th column of $\mathbf{Corr}$ and $\mathbf{P}_{e,k}$, respectively.

The forecast step propagates the ensemble from time $t_k$ to $t_{k+1}$ with the nonlinear model $\mathcal{M}_{k,k+1}$. For the analysis step we use the efficient Ensemble Transform Kalman Filter (ETKF) introduced by Bishop et al. (2001) and follow the implementation of the algorithm given by Hunt et al. (2007).

The ETKF may generate ensembles of analyses with underestimated spread, which can lead to the divergence of the filter. We use an inflation procedure (Anderson and Anderson, 1999) here to avoid this potential degeneracy. In the rest of the paper
the inflation factor is denoted by the parameter $\lambda_{\mathrm{infla}}$.

In the twin experiments performed in Sect. 4 and 5 we use a large number of ensembles to avoid producing spurious correlations in $\mathbf{P}_{e,k}$. Therefore, no localisation has been employed in this paper.

### 3.2   Form of the state vector in the moving mesh case

Traditionally, in a data assimilation scheme the state vector includes all the physical variables of the given dynamical system.
For a fixed-grid numerical method the state variables are defined at fixed spatial positions. For example, for a grounded ice sheet modelled with a fixed-grid method (and assuming every parameter is perfectly known), the unknown variables are the ice thicknesses located at known positions (see e.g. Bonan et al., 2014).

In contrast, the primary characteristic of a moving-point method is that the numerical domain evolves in time. The positions of the nodes evolve jointly with the state variables according to the dynamical system equations and can be updated using the
assimilation scheme. We therefore include the positions of the points in the state vector. As a consequence we define the state vector $\mathbf{x}$ as follows

$$\mathbf{x} = \begin{pmatrix} \mathbf{x}_h \\ \mathbf{x}_r \end{pmatrix} \qquad \text{with} \qquad \mathbf{x}_h = \begin{pmatrix} h_1 \\ \vdots \\ h_{n_r-1} \end{pmatrix} \qquad \text{and} \qquad \mathbf{x}_r = \begin{pmatrix} \hat{r}_2 \\ \vdots \\ \hat{r}_{n_r} \end{pmatrix}. \tag{26}$$

Estimates obtained by combining DA with this formulation of $\mathbf{x}$ using a moving-point numerical model provide more information on the state of the system than if we were using a fixed-grid method.

In particular, for an ice sheet model, this approach gives us a direct estimation of the position of the ice sheet margin that cannot be obtained in fixed-grid methods without interpolation. In this case, we do not include in $\mathbf{x}$ the ice thickness at the margin $h_{n_r}$ or the position of the ice divide $\hat{r}_1$ as both are fixed to zero. The DA schemes must, however, provide estimates with strictly positive ice thicknesses $h_i$, $i = 1, \ldots, n_r - 1$, and a preserved order for node positions to respect the assumption of the moving mesh scheme.

This can be achieved with 3D-Var if the specified background covariance matrix $\mathbf{B}_k$ in Eq. (21) is prescribed carefully. At time $t_k$ we decompose the background error covariance matrix $\mathbf{B}$ and the tangent linear matrix of the observation operator $\mathbf{H}$



(we drop the time index $k$ for clarity) as

$$\mathbf{B} = \begin{pmatrix} \mathbf{B}_h & \mathbf{B}_{rh}^T \\ \mathbf{B}_{rh} & \mathbf{B}_r \end{pmatrix} \quad \text{and} \quad \mathbf{H} = \begin{pmatrix} \mathbf{H}_h & \mathbf{H}_r \end{pmatrix} = \begin{pmatrix} \frac{\partial \mathcal{H}}{\partial \mathbf{x}_h}(\mathbf{x}^f) & \frac{\partial \mathcal{H}}{\partial \mathbf{x}_r}(\mathbf{x}^f) \end{pmatrix}, \tag{27}$$

where $\mathbf{B}_h$ is the background error covariance matrix between the state variables, $\mathbf{B}_r$ is the error covariance between mesh point locations and $\mathbf{B}_{rh}$ includes the cross-covariances between errors in point locations and errors in state variables. The different components of the state vector are then updated by the following analysis step

$$\mathbf{x}_h^a = \mathbf{x}_h^b + \left( \mathbf{B}_h \mathbf{H}_h^T + \mathbf{B}_{rh}^T \mathbf{H}_r^T \right) \left( \mathbf{HBH}^T + \mathbf{R} \right)^{-1} \left( \mathbf{y} - \mathcal{H}\left( \mathbf{x}^b \right) \right) \tag{28}$$

$$\mathbf{x}_r^a = \mathbf{x}_r^b + \left( \mathbf{B}_{rh} \mathbf{H}_h^T + \mathbf{B}_r \mathbf{H}_r^T \right) \left( \mathbf{HBH}^T + \mathbf{R} \right)^{-1} \left( \mathbf{y} - \mathcal{H}\left( \mathbf{x}^b \right) \right) \tag{29}$$

The most difficult step with this form of analysis is, in general, to set appropriately the cross-covariances in $\mathbf{B}_{rh}$ that are needed for the update stage. For example, if either $\mathbf{H}_h$ or $\mathbf{H}_r$ is zero, a non-zero $\mathbf{B}_{rh}$ matrix is the only way to correct estimates of both $\mathbf{x}_h$ and $\mathbf{x}_r$. However, we will see in the next section that in our assimilation systems for the ice sheet model, the observation operator depends explicitly on both ice thickness variables and mesh node locations and, therefore, by setting $\mathbf{B}_{rh}$ to zero we can still obtain good estimates.

For the moving-point ice sheet model, the DA analysis step updates both ice thickness variables and node positions, but the total mass and mass fractions have to be updated as well, since they are not preserved by the analysis (and there is no reason to preserve them). Therefore these quantities need to be 'reset' from the analysed state vector. This is easily done by using Eq. (14) and Eq. (15). The adapted 3D-Var scheme is performed according to the following steps:

1. Calculate a forecast of the state vector $\mathbf{x}^b$ by using the previous analysis solution to initialise the numerical moving point model.

2. Use the analysis scheme (Eq. (28) and Eq. (29)) to produce the analysis $\mathbf{x}^a$.

3. From $\mathbf{x}^a$, calculate the analysed total mass $\theta^a$ and update the mass fractions $\mu^a$ using Eq. (14) and Eq. (15).

4. Evolve the analysis solution using the numerical moving point model to the next time where observations are available.

5. Repeat steps 2–5.

The adapted ETKF roughly follows the same path as 3D-Var except that, at step 1, we calculate the forecast for each member of the ensemble and, at step 3, the total mass and mass fractions have to be updated for each member of the ensemble (they are different for each ensemble member). The strict positivity of ice thickness variables and the order required in Eq. (13) for node positions are ensured by appropriately setting the initial ensemble in the ETKF.

### 3.3 Type of observations assimilated

In the twin experiments performed in Sect. 4 and 5, we use three different conventional types of observations of an ice sheet system that are available in reality (see e.g. Vaughan et al., 2013). The first is direct observations of the ice thickness. Assuming





that we have an observation of the ice thickness located at position $r^o$, we define the associated observation operator as

$$\mathcal{H}(\mathbf{x}) = \begin{cases} h_i + \dfrac{r^o - \hat{r}_i}{\hat{r}_{i+1} - \hat{r}_i} \left(h_{i+1} - h_i\right) & \text{if } \hat{r}_i \leq r^o \leq \hat{r}_{i+1} \\ 0 & \text{elsewhere}, \end{cases} \tag{30}$$

which is merely a piecewise linear interpolation operator. Note that $\mathcal{H}$ depends on both ice thickness variables $h_i$ and node locations $\hat{r}_i$. We also assimilate observations of surface elevation and surface ice velocity. We again use a piecewise linear

interpolation operator as in Eq. (30). For observations of surface elevation, we have

$$\mathcal{H}(\mathbf{x}) = \begin{cases} s_i + \dfrac{r^o - \hat{r}_i}{\hat{r}_{i+1} - \hat{r}_i} \left(s_{i+1} - s_i\right) & \text{if } \hat{r}_i \leq r^o \leq \hat{r}_{i+1} \\ b(r_i) & \text{elsewhere} \end{cases} \tag{31}$$

with

$$s_i = h_i + b(r_i). \tag{32}$$

For observations of surface ice velocity, we have

$$\mathcal{H}(\mathbf{x}) = \begin{cases} u_{s,i} + \dfrac{r^o - \hat{r}_i}{\hat{r}_{i+1} - \hat{r}_i} \left(u_{s,i+1} - u_{s,i}\right) & \text{if } \hat{r}_i \leq r^o \leq \hat{r}_{i+1} \\ 0 & \text{elsewhere} \end{cases} \tag{33}$$

with

$$u_{s,i} = \frac{1}{2} A \left(\rho_i g\right)^3 \operatorname{sgn}\left(s_{n_r} - s_{n_r-1}\right) \left| h_i^4 \left(\frac{\partial b}{\partial r}(r_i)\right)^3 + \frac{3}{5} \frac{h_i^5 - h_{i-1}^5}{\hat{r}_i - \hat{r}_{i-1}} \left(\frac{\partial b}{\partial r}(r_i)\right)^2 \right. \tag{34}$$

$$\left. + \frac{1}{3} \left(\frac{h_i^3 - h_{i-1}^3}{\hat{r}_i - \hat{r}_{i-1}}\right)^2 \frac{\partial b}{\partial r}(r_i) + \frac{27}{343} \left(\frac{h_i^{7/3} - h_{i-1}^{7/3}}{\hat{r}_i - \hat{r}_{i-1}}\right)^3 \right|, \tag{35}$$

except for $u_{s,1} = 0$.

We may also assimilate observations of the position of the ice sheet margin. Using a moving point method allows the movement of boundaries to be tracked explicitly. In our context, the position of the ice sheet margin is represented by $\hat{r}_{n_r}$. As a consequence the observation operator for such an observation is defined by

$$\mathcal{H}(\mathbf{x}) = \hat{r}_{n_r}. \tag{36}$$

The operator is continuous and linear. This makes the assimilation of the position of the margin straightforward in comparison

with the same assimilation with a fixed grid model (see e.g. Lecavalier et al., 2014).





## 4 Numerical experiments with an idealized model

To demonstrate the efficiency of our DA approach, we perform twin experiments with two different configurations. In this section we consider experiments using an idealized system with a flat bedrock and the EISMINT surface mass balance detailed in Eq. (A1).

5 ### 4.1 Experimental design

We first generate a model run with the moving point numerical model from known initial conditions. From this simulation observations are created with added error sampled from a Gaussian distribution. This run is used as a reference to measure the quality of the DA estimates.

We define the reference initial ice thickness profile by the function,

$$10 \quad h(0,r) = h_0 \left( 1 - \left( \frac{r}{r_{\max}} \right)^2 \right)^{3/7} \qquad 0 \leq r \leq r_{\max} \tag{37}$$

where $h_0 = 2000$ m and $r_{\max} = 450$ km. This function gives a smooth interior profile with a steep snout at the ice sheet margin $r_{\max}$. This is in compliance with the physics involved in the ice sheet model and provides an initial state with a margin that is immediately in motion. The reference run is obtained with an initial mesh of $n_r = 28$ points evenly spaced between $\hat{r}_1 = 0$ and $\hat{r}_{n_r} = 450$ km. The model time step is $\Delta t = 0.02$ yr, the bed elevation $b$ is fixed to zero and the surface mass balance used is

15 from the EISMINT benchmark (Eq. (A1)). The experiment starts at time $t = 0$ yr and ends at $t = 2000$ yr. The evolution of the reference ice thickness profile can be seen in Fig. 2.

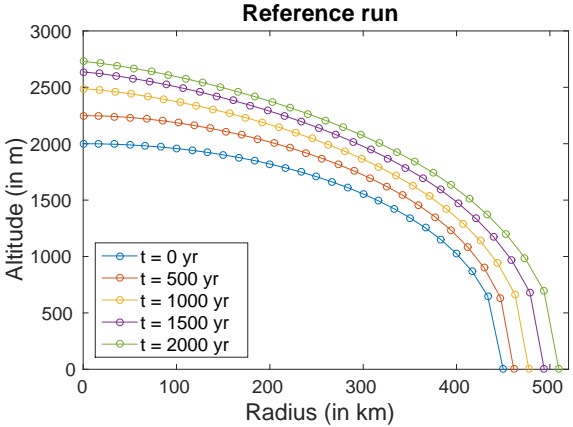

**Figure 2.** Ice thickness profile from the reference run in a simple case (flat bedrock, EISMINT surface mass balance from Eq. (A1)). The initial state follows the profile of Eq. (37) with $h_0 = 2000$ m and $r_{\max} = 450$ km. The reference run is obtained with an initial mesh of $n_r = 28$ points evenly spaced between $\hat{r}_1 = 0$ and $\hat{r}_{n_r} = 450$ km.

From the reference run we generate observations of ice thickness and the position of the ice sheet margin at times $t_1 = 500$ yr and $t_2 = 1500$ yr. Observations of thickness are taken at each point except at the margin (so a total of 27 observations) with



added random noise from the Gaussian distribution $\mathcal{N}(0, \sigma_h^{o\,2})$, $\sigma_h^o = 100$ m. For the position of the margin, the observational noise is sampled from $\mathcal{N}(0, \sigma_r^{o\,2})$, $\sigma_r^o = 10$ km.

To evaluate the performance of our DA approaches, we compare the estimated ice thickness profiles with their reference counterparts. This is mostly done graphically. We also study the quality of the estimates of two variables: the ice thickness at the ice divide $r = 0$ and the position of the ice sheet margin.

## 4.2 Updating the ice thickness only

We begin by studying the performance of the DA schemes in the idealized configuration where we assimilate observations of ice thickness only. We start with an experiment using the 3D-Var algorithm in which only the ice thickness is updated at the assimilation times and the mesh point positions are not updated.

The background state is defined as follows:

- at initial time, the background ice thickness profile is set using the same profile as the reference (Eq. (37)) but with $h_0 = 2100$ m ($+5$ % error from the reference) and $r_{\max} = 472.5$ km (also $+5$ % error),

- the background mesh consists of $n_r = 28$ points, evenly spaced between $\hat{r}_1 = 0$ and $\hat{r}_{n_r} = 472.5$ km at initial time,

- the model time step is $\Delta t = 0.02$ yr.

As we are using a 3D-Var scheme in this experiment, the background error covariance matrix $\mathbf{B}$ needs to be prescribed at both times of assimilation ($t_1 = 500$ yr and $t_2 = 1500$ yr). In this first experiment we only update ice thickness variables so we set the background error covariance matrix for point positions $\mathbf{B}_r$ and the cross-covariance matrix $\mathbf{B}_{rh}$ to zero. We define $\mathbf{B}_h$ the covariance matrix for ice thickness variables as

$$\mathbf{B}_h = \mathbf{D}_h^{1/2} \mathbf{C}_h \mathbf{D}_h^{1/2} \tag{38}$$

with $\mathbf{D}_h$ the diagonal variance matrix and $\mathbf{C}_h$ the correlation matrix. $\mathbf{D}_h$ is simply set to $\sigma_h^{b\,2} \mathbf{I}_{n_r-1}$ with $\sigma_h^b = 100$ m. The background error correlation structure follows a Second-Order AutoRegressive (SOAR) distribution with

$$[\mathbf{C}_h]_{i,j} = \left(1 + \frac{|\hat{r}_i^b - \hat{r}_j^b|}{L_h}\right) \exp\left(-\frac{|\hat{r}_i^b - \hat{r}_j^b|}{L_h}\right) \qquad i,j = 1, \ldots, n_r - 1, \tag{39}$$

where $[\mathbf{C}_h]_{i,j}$ denotes the entry in the $i$-th row and $j$-th column of $\mathbf{C}_h$, $\hat{r}_i^b$ the location of the $i$-th mesh point of the background state at the time of assimilation and $L_h$ is some correlation length scale to be fixed. The SOAR function is preferred to a Gaussian structure as the matrix $\mathbf{C}_h$ is better conditioned for inversion in that case (Haben et al., 2011). We set $L_h$ to 100 km.

This definition of $\mathbf{B}$ takes into account the flow dependency of the moving point locations, making our approach adaptive. Figure 3 displays $\mathbf{B}_h$ at assimilation times $t_1 = 500$ yr and $t_2 = 1500$ yr. As the distance between grid points increases in time in the experiment, the covariances tend to reduce between the two assimilation times. For example the covariance between the location of points $\hat{r}_1^b$ and $\hat{r}_{n_r-1}^b$ is reduced from $[\mathbf{B}_h]_{1,n_r-1} = 530.7$ at $t_1 = 500$ yr to $[\mathbf{B}_h]_{1,n_r-1} = 446.6$ at $t_2 = 1500$ yr. In





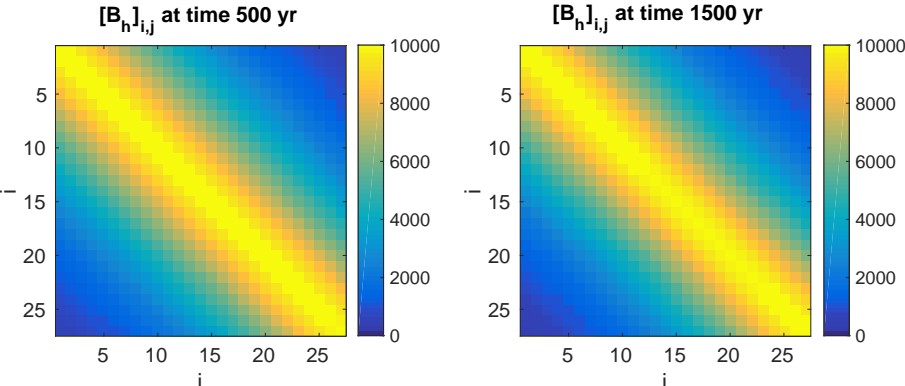

**Figure 3.** Covariance matrices for ice thickness variables $\mathbf{B}_h$ used by the 3D-Var at assimilation times $t_1 = 500$ yr and $t_2 = 1500$ yr. Covariances between variables at distant locations tend to reduce between the two assimilation times. The distance between adjacent nodes also tends to be greater in the centre of the mesh than at the boundaries, leading to a decreasing covariance at $t_2 = 1500$ yr in this area.

addition we note decreased correlations for points around the centre of the mesh due to a greater distance between adjacent nodes in the centre of the grid than at the boundaries.

The formulation of $\mathbf{B}$ forces the re-computation of the matrix at every assimilation time. This is a limiting factor of our 3D-Var approach, especially for high dimensional systems making it cost more than traditional 3D-Var for fixed-grid models
5   in which $\mathbf{B}$ is only computed once. Nevertheless, this approach ensures that this moving-point framework produces positive estimates of ice thickness variables and a smooth interior profile in accordance with the physics of the system.

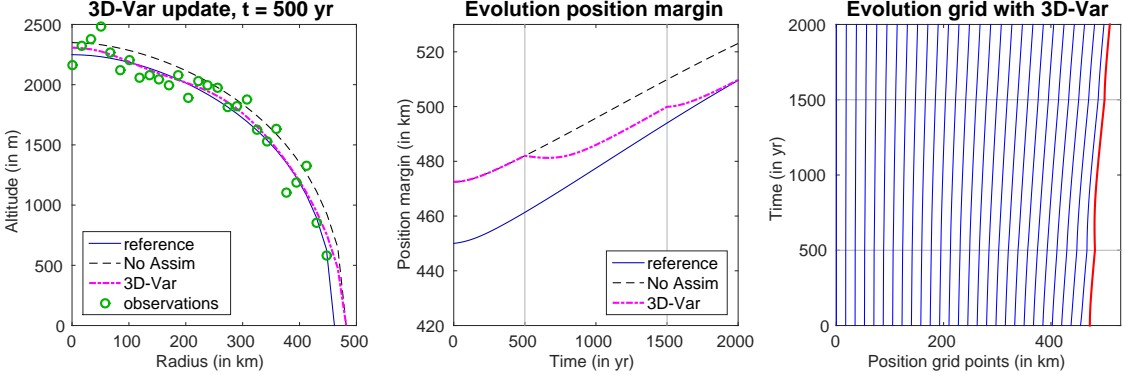

**Figure 4.** *Left:* 3D-Var analysis at time $t = 500$ yr compared with the forecast and the reference when we update only ice thickness variables. The ice thickness profile is improved, especially between $r = 100$ km and $r = 400$ km. *Centre:* Evolution of the position of the margin with time. Even if the position of the margin is not directly updated, the trajectory of the margin is corrected as a result of the ice thickness update. *Right:* Evolution of the position of grid points with time. The trajectory of each grid node is corrected after each analysis, as is the margin.




We now evaluate the quality of the estimates. Fig. 4 (left) displays the analysed ice thickness profile compared to its background and reference counterparts at the first time of assimilation $t_1 = 500$ yr. The picture shows that the ice thickness profile in the interior of the ice sheet is substantially improved by DA. For example the absolute error in ice thickness at the ice divide ($r = 0$) is reduced from 100 m to 58.3 m by the 3D-Var analysis. Results are even better between $r = 100$ km and 400 km.

Since we only update $\mathbf{x}_h$ in this experiment, the position of the margin is not modified by our update. Nevertheless, by correcting the interior of the ice sheet, the forecast of the migration of the margin is improved (see central and right picture after $t = 500$ yr, Fig. 4), and at the second assimilation time, $t = 1500$ yr, the absolute difference between the position of the margin before analysis and its reference position is only 5.6 km (instead of 15.9 km without DA).

### 4.3 Updating ice thickness variables and node positions

We now use 3D-Var to update both ice thickness variables and node locations. The definitions of $\mathbf{B}_h$ and $\mathbf{B}_{rh}$ remain the same as in the previous experiment, but we set the covariance matrix for node positions $\mathbf{B}_r$ to be $\mathbf{B}_r = \mathbf{D}_r^{1/2}\mathbf{C}_r\mathbf{D}_r^{1/2}$ with $\mathbf{D}_r$ the diagonal variance matrix and $\mathbf{C}_r$ a correlation matrix. The matrix $\mathbf{D}_r$ is set to to $\sigma_r^{b\,2}\mathbf{I}_{n_r-1}$ with $\sigma_r^b = 22.5$ km and $\mathbf{C}_r$ follows a SOAR distribution with

$$[\mathbf{C}_r]_{i,j} = \left(1 + \frac{|\hat{r}_{i+1}^b - \hat{r}_{j+1}^b|}{L_r}\right)\exp\left(-\frac{|\hat{r}_{i+1}^b - \hat{r}_{j+1}^b|}{L_r}\right), \qquad i,j = 1,\ldots,n_r-1, \tag{40}$$

where $L_r$ is a correlation length scale fixed to 100 km. The formulation of $\mathbf{B}_r$ aims to ensure that the order between mesh points defined by Eq. (13) is preserved by the 3D-Var algorithm. Since the distance between nodes evolves in time, it is even more important than in the previous case to use a flow-dependent background error covariance matrix $\mathbf{B}$.

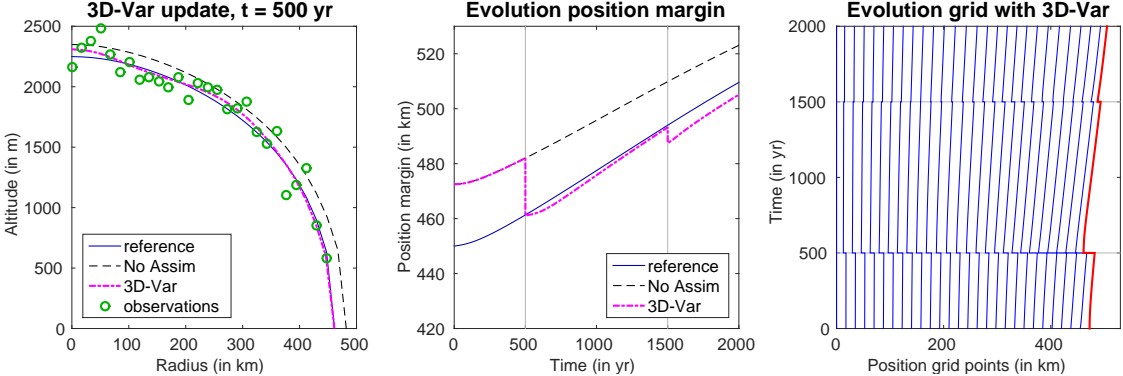

**Figure 5.** *Left:* 3D-Var analysis at time $t = 500$ yr compared with the forecast and the reference when we update ice thickness variables and node locations. In contrast to the results shown in Fig. 3, the ice thickness profile is substantially improved close to the margin. *Centre:* Evolution of the position of the margin with time. The estimates are of very good quality even if the margin is not observed directly. *Right:* Evolution of the position of mesh points with time. The trajectory of each node is corrected by each analysis, as is the margin.

Results for the ice thickness profile are shown in Fig. 5. Overall estimates obtained with updating both ice thickness variables and node positions are better than when we update only ice thickness variables. The absolute error in ice thickness at the ice





divide ($r = 0$) is reduced from 100 m to 60.2 m by the 3D-Var analysis at time $t_1 = 500$ yr, which is similar to the previous experiment. However, we now obtain at $t_1 = 500$ yr a very accurate ice thickness profile close to the margin and its estimated position has an absolute error of only 0.2 km. This shows that the estimated position of the ice sheet margin can be accurately corrected by only using standard observations (no observation of the position of the margin is involved in this experiment). At the second time of assimilation at $t_2 = 1500$ yr, the estimate is degraded, however, as a result of using fixed variances in the matrix $\mathbf{B}$. This behaviour is discussed further in Sect. 4.5.

In these experiments we have specified a fixed form for the the background error covariance matrices. We next show, using an ETKF, how the covariances are expected to evolve with time and the effects of this on the assimilation.

### 4.4 Using an ETKF

We now perform the same experiment as before except that we now use an ETKF. The key question is how to generate the initial ensemble composed of $N_e$ members. The easiest way is to add noise to a background state sampled from a Gaussian law $\mathcal{N}(\mathbf{0}, \mathbf{B})$ with $\mathbf{B}$ the background error covariance matrix defined in Eq. (27).

In this experiment we generate an initial ensemble of $N_e = 200$ members using:

- the same background state used in the experiments detailed in Sect. 4.2 and 4.3,

- $\mathbf{B}_h$ defined by Eq. (38) with the diagonal matrix $\mathbf{D}_h = \sigma_h^2 \mathbf{I}_{n_r-1}$, $\sigma_h^b = 100$ m, $\mathbf{C}_h$ defined by Eq. (39), $L_h = 100$ km,

- $\mathbf{B}_r$ taken as $\mathbf{D}_r^{1/2} \mathbf{C}_r \mathbf{D}_r^{1/2}$ with $\mathbf{C}_r$ defined by Eq. (40) with $L_h = 100$ km and the diagonal matrix $\mathbf{D}_r$ defined as

$$[\mathbf{D}_r]_{ii} = \min\left(\sigma_r^b, \alpha\,\hat{r}_i\right) \qquad i = 1, \ldots, n_r - 1 \tag{41}$$

with $\sigma_r^b = 22.5$ km and $\alpha = 0.2$,

- $\mathbf{B}_{rh}$ set to zero.

Note that the definition of $\mathbf{B}$ is slightly different from the previous experiment as we choose different diagonal variances. The change is because of the high probability of generating useless initial meshes with negative radii using $\mathbf{D}_r = \sigma_r^{b\,2} \mathbf{I}_{n_r-1}$ as the background standard deviation $\sigma_r^b$ is larger than the background position of the first points (for example $\hat{r}_2^b = 17.5$ km). To avoid this problem we have decided just to reduce the variance for the position of points near the ice divide using Eq. (41). The new ensemble mean has, at the initial time, an estimated position of the margin of 472.9 km with an estimated standard deviation of 22.8 km (where the true value at $t = 0$ a is 450 km).

We do not use any inflation in this experiment ($\lambda_{\mathrm{infla}} = 1$).

Results are summarised in Fig. 6. At the first time of assimilation $t_1 = 500$ yr, the analysis step corrects the ice thickness profile well. The estimate of the ice thickness at $r = 0$ is of the same quality as in the previous experiments (absolute error of 46.9 m) and the estimate of the position of the margin is reduced from 483.1 km (forecast mean with estimated standard deviation 18.9 km) to 468.8 km (analysis mean with estimated standard deviation 7.1 km). The estimate obtained by the ETKF is in accordance with the true value (which is within the ensemble spread) and the absolute error of 7.5 km is of the same




order as the estimated standard deviation. The rest of the experiment exhibits the same quality in terms of recovering the ice thickness profile.

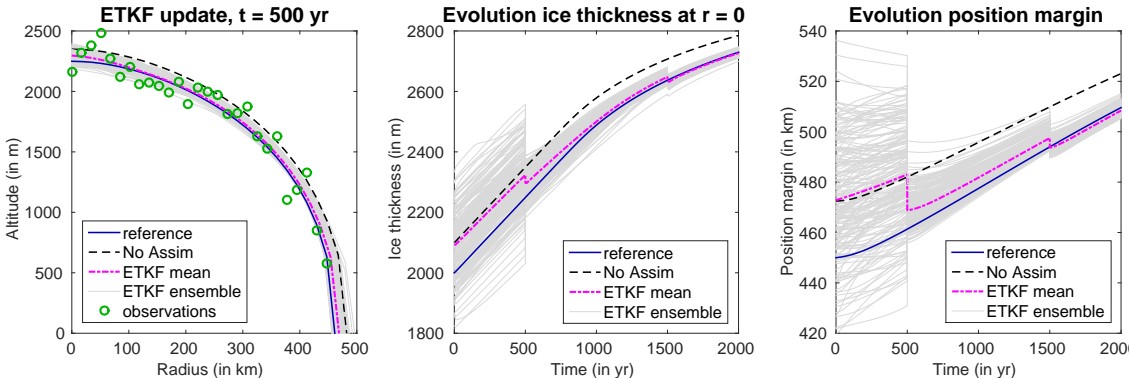

**Figure 6.** *Left:* ETKF analysis at time $t = 500$ yr compared with the forecast and the reference. The ice thickness profile is improved over the whole domain and the reference profile is within the ensemble spread. *Centre:* Evolution of the ice thickness at $r = 0$ with time. The estimates are of very good quality and the estimates seem to converge towards the reference value at the end of the study. *Right:* Evolution of the position of the margin with time. The ETKF provides consistent estimates and the reference value is always within the ensemble spread.

The ETKF provides information on the covariance structures for ice thickness variables and mesh point positions. We display estimated standard deviations and an estimate of the correlation matrix **Corr** (see Eq. (25)) in Fig. 7 for the analysis ensemble
5   at time $t = 500$ yr.

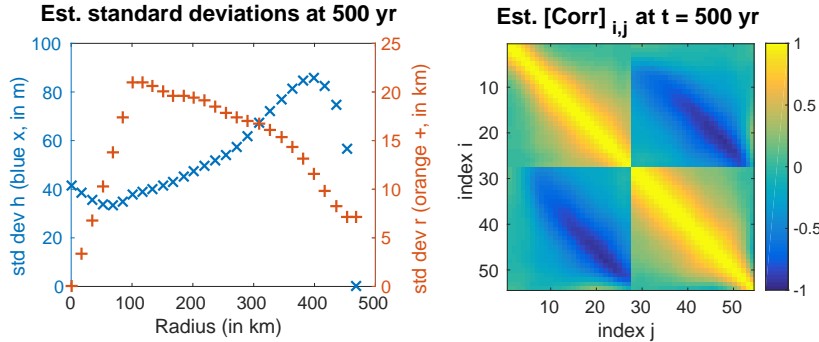

**Figure 7.** Standard deviations and correlation matrix **Corr** estimated from the ETKF analysis ensemble at time $t = 500$ yr when we use only observations of ice thickness. Auto-correlations between ice thicknesses are located in the top left corner of **Corr**, auto-correlations between node positions in the bottom right corner. The rest of the matrix depicts the cross-correlations.

The ETKF produces decreased standard deviations and correlation length scales for ice thickness variables close to the ice divide. For example the standard deviation of the ice thickness at the ice divide is more than halved by the analysis, from $97.4$ m before analysis to $41.6$ m. Decreased standard deviations and correlation length scales are also obtained for node locations but




close to the margin in this case. The standard deviation for the position of the margin is reduced from 18.9 km to 7.1 km by the analysis. The ETKF also produces strong anti-correlations between ice thickness variables and node positions, meaning that where ice thickness variables become larger associated nodes need to retreat to fit the observations of ice thickness.

### 4.5  Comparing 3D-Var and the ETKF

We now compare the results from applying the 3D-Var and ETKF assimilation schemes in the case where we observe only the ice thickness. We focus on the accuracy of the estimated ice thickness at $r = 0$ and the position of the margin.

Figure 8 shows the evolution of the absolute errors in the estimates of the ice thickness at $r = 0$ and in the position of the margin for the ETKF and for 3D-Var, with and without node updates. All three methods provide improved estimates at the first analysis time ($t_1 = 500$ yr), leading to good forecasts up to the next assimilation time. We find that the ETKF tends to perform

better than the variational approach and that for 3D-Var the estimates obtained by updating both ice thickness variables and node positions are generally better than those where only ice thickness variables are updated.

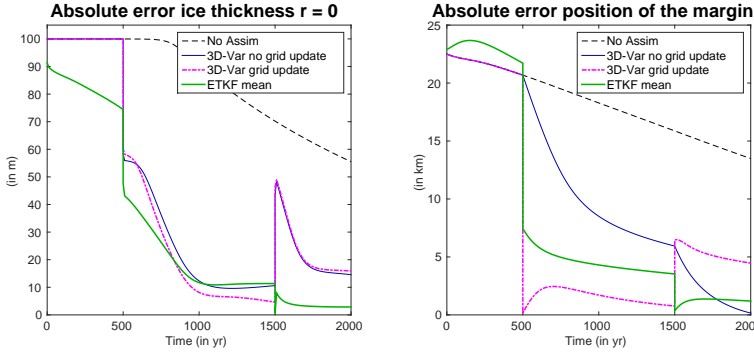

**Figure 8.** Evolution of the absolute error of the estimated ice thickness at $r = 0$ and the estimated position of the margin when we observe only the ice thickness. We compare the absolute errors obtained when we use 3D-Var without and with correction of the position of grid nodes and when we use an ETKF.

For 3D-Var without node updates, the analysis at the second time of assimilation ($t_2 = 1500$ yr) of the ice thickness at $r = 0$ is unfortunately degraded relative to the forecast, but the estimated position of the margin is still improved by the second analysis. In the case where ice thickness and nodes are updated, the estimates of both ice thickness at $r = 0$ and the position of

the margin are degraded at the second time of assimilation. This weakens the confidence in the forecast and we partially lose what we had gained from the previous analysis. This effect would not necessarily appear with another set of observations of thickness, but the experiment shows the sensitivity of 3D-Var to current observations because of the use of fixed variances in the prescribed covariance matrix $\mathbf{B}$.

Using the ETKF assimilation scheme, where the covariance matrix fully evolves in time, is seen to improve the overall

estimates. At each assimilation time, the errors in the estimated ice thickness and the position of the margin are decreased. Notably we do not observe any degrading of the estimates at the second time of assimilation. This demonstrates the better





ability of the ETKF to provide accurate estimates in the context of the moving point model. Remembering past observations through the ensemble ensures that the ETKF is a more reliable scheme than 3D-Var. This improvement has a computational cost, however, as we now need to run the model $N_e$ times instead of once for 3D-Var.

### 4.6 Assimilating observations of the position of the margin

In this section we perform the same experiments as previously, but we now assimilate not only the same observations of ice thickness as before but also observations of the position of the margin. We consider only the case of 3D-Var with grid update and the ETKF.

Absolute errors for the estimates of the ice thickness at $r = 0$ and the position of the margin are shown in Fig. 9. In both cases assimilating observations of the position of the margin is beneficial to our estimates of the margin and of the ice thickness profile close to the margin. For example the estimated position of the margin at time t = 500 yr has an absolute error of $4.2$ km for the ETKF (compared to 7.5 km previously). Not surprisingly it does not change the results for the ice thickness at $r = 0$.

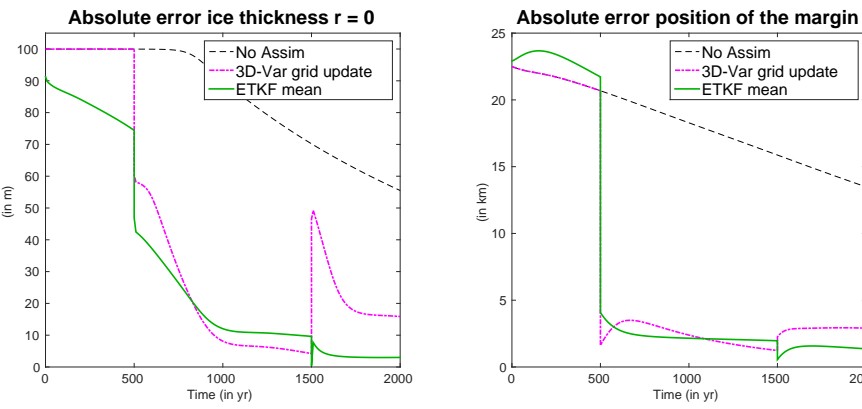

**Figure 9.** Evolution of the absolute error of the estimated ice thickness at $r = 0$ and the estimated position of the margin when we observe the ice thickness and the position of the margin. We compare the absolute errors obtained when we use 3D-Var with correction of the position of grid nodes and when we use an ETKF. In both experiments the results are improved with respect to the position of the margin (compared to results detailed in Fig. 8). No improvement (nor degradation) is observed for the ice thickness at $r = 0$.

Adding observations of the position of the margin in the data assimilation system reduces the estimated standard deviations obtained with the ETKF for variables close to the margin. For example, the estimated standard deviation for the position of the margin is now $5.8$ km instead of $7.1$ km. Not surprisingly it has no influence on the standard deviation for variables close to the ice divide. The estimated correlation structure (not shown) is also not modified by adding observations of the position of the margin in the DA system.



## 5  Numerical experiments with an advanced configuration

In this section we consider experiments using a more realistic configuration with a non-flat bedrock and an advanced surface mass balance, detailed in Appendix A2.

### 5.1  Experimental Design

We generate observations from a new reference run. We use a non flat fixed bedrock whose elevation is defined by the equation

$$b(r) = 1000\,\text{m} - 1400\,\text{m} \cdot \left(\frac{r}{1000\,\text{km}}\right)^2 + 700\,\text{m} \cdot \left(\frac{r}{1000\,\text{km}}\right)^4 - 120\,\text{m} \cdot \left(\frac{r}{1000\,\text{km}}\right)^6 \tag{42}$$

The reference run is generated from a realistic initial state obtained with the following steps:

- Start with an ice sheet profile following eq. (37) with $h_0 = 2000$ m, $r_{\max} = 300$ km and $n_r = 21$ computational mesh points evenly spaced between $\hat{r}_1 = 0$ and $\hat{r}_{n_r} = 300$ km.

- Run the numerical model with a fixed climate forcing where $T_{\text{clim}} = 4\,°\text{C}$ until reaching the steady state (a $30{,}000$ yr run with a $\Delta t = 0.01$ yr time step).

- From this steady state, run the numerical model with a linearly warming climate forcing from $T_{\text{clim}} = 4\,°\text{C}$ with $dT_{\text{clim}}/dt = 0.02\,°\text{C.yr}^{-1}$ for an extra 100 yr ($\Delta t = 0.01$ yr). The state obtained at the end of the run is the initial state (see Fig. 10).

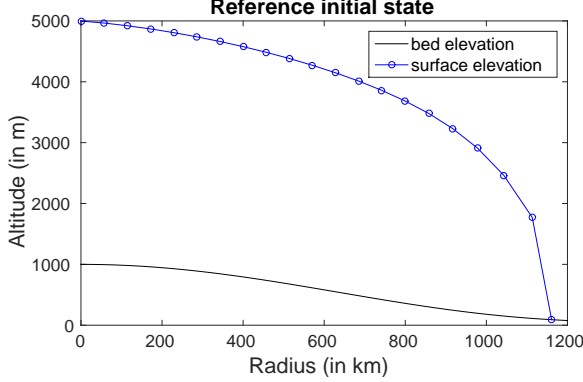

**Figure 10.** Initial state used to obtain a 20-year reference run under a warming climate as detailed in Sect. 5.1 with $n_r = 21$ grid points and a non-flat bed.

The reference is obtained by running the model over 20 years from the initial state with a time step $\Delta t = 0.01$ yr and the same linearly warming climate forcing (with $T_{\text{clim}} = 6\,°\text{C}$ at initial time $t = 0$ yr, and $T_{\text{clim}} = 6.4\,°\text{C}$ at $t = 20$ yr). Over the 20-year run, the geometry of the ice sheet stays relatively similar to the geometry of the initial state due to the slow dynamics





of the model. The ice sheet margin retreats from 1160.9 km to 1158.6 km and the ice thickness at the ice divide increases by 1.5 m.

We generate observations of surface elevation, surface ice velocity and the position of the ice sheet margin at times $t = 1, 2, \ldots, 10$ yr from the reference run. The observations of the surface are taken at each point including the margin with

an added Gaussian noise (uncorrelated with standard deviation $\sigma_s^o = 200$ m). The observations of the surface ice velocity are located at the mid-points between mesh points (so we have 20 observations of surface velocity). Observations are noised using a Gaussian law (standard deviation $\sigma_{u_s}^o = 30 \, \mathrm{m.yr}^{-1}$, uncorrelated). For the position of the margin, the observational noise is sampled from $\mathcal{N}(0, \sigma_r^{o2})$ with $\sigma_r^o = 50$ km.

We compare the influence of the observations on the quality of the DA estimates and the subsequent forecasts for the 3D-Var

and ETKF methods. Again we focus on the two variables: the ice thickness at the ice divide $r = 0$ and the position of the ice sheet margin.

### 5.2 Assimilating observations of surface elevation

We begin by studying the performance of the DA schemes where we assimilate only observations of surface elevations.

For 3D-Var the estimates are obtained using an initial background state defined as $\mathbf{x}^b = 0.95 \, \mathbf{x}^{\mathrm{ref}}(0)$ with a 5% smaller extent

than the reference state. The flow-dependent background error covariance matrix $\mathbf{B}$ is defined as in Eq. (27). The matrix $\mathbf{B}_h$ is defined as in Eq. (38) with a SOAR matrix for $\mathbf{C}_h$ ($\sigma_h^b = 200$ m, $L_h = 240$ km) and $\mathbf{B}_r$ is defined with a SOAR matrix for $\mathbf{C}_r$ ($\sigma_r^b = 60$ km, $L_r = 240$ km). The matrix $\mathbf{B}_{rh}$ is set to $\mathbf{0}$.

The ETKF uses an ensemble with 200 members. The initial ensemble is generated by adding to $\mathbf{x}^b$ a random noise drawn from the Gaussian law $\mathcal{N}(\mathbf{0}, \mathbf{B})$. The background covariance matrix $\mathbf{B}$ is defined as previously, except for $\mathbf{B}_r$ for which we

still use a SOAR matrix for $\mathbf{C}_r$ ($L_r = 240$ km) but with variances decreased near the ice divide following Eq. (41) ($\sigma_r^b = 60$ km and $\alpha = 0.2$). We tested different values for the inflation parameter $\lambda_{\mathrm{infla}}$; the best results were obtained with $\lambda_{\mathrm{infla}} = 1.01$.

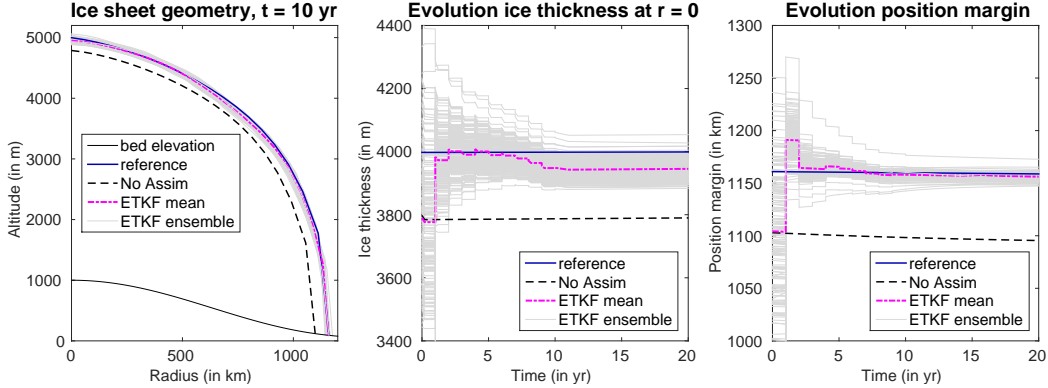

**Figure 11.** ETKF results for the advanced configuration where observations of surface elevation are assimilated over the first 10 years and a forecast is made for 10 further years. *Left:* ETKF analysis at time $t = 10$ yr compared with the reference. *Centre:* Evolution of the ice thickness at $r = 0$ with time. *Right:* Evolution of the position of the margin with time.





We first study the results obtained with the ETKF. At the end of the data assimilation window, $t = 10$ yr, the ice thickness profile is retrieved well everywhere by the mean of the ensemble and the reference profile is within the ensemble spread (see Fig. 11). We note that the estimate of the ice thickness at the ice divide is improved by the first analysis. After time $t = 7$ yr, however, the estimate is worsened by the analysis. This is because the ensemble spread is too small in that area. This can be

fixed by taking a larger inflation parameter $\lambda_{\text{infla}}$, but the estimates of other variables are then degraded. The estimated position (mean) of the margin at $t = 10$ yr is 1158.0 km with an ensemble standard deviation of 3.1 km. In comparison to the reference value at that time, $r = 1159.9$ km, we see that the ETKF with a large ensemble performs well. The quality of the estimates is also kept high during the forecast (from $t = 10$ yr to $t = 20$ yr). For example the absolute error on the position of the margin is kept below 2.5 km over this time window.

With respect to the covariance matrix, the estimates seem to show a similar behaviour to those of the experiment detailed in Sect. 4.4 using the ETKF where observations of ice thickness are assimilated (see Fig. 12), but with a larger correlation length scale. The similarity can be explained by the similarity of the construction of the initial ensemble (the same structure for the background covariance matrix $\mathbf{B}$ used to sample the Gaussian noise added to the background state) and by the similarity of the observation operators for ice thickness and surface elevation.

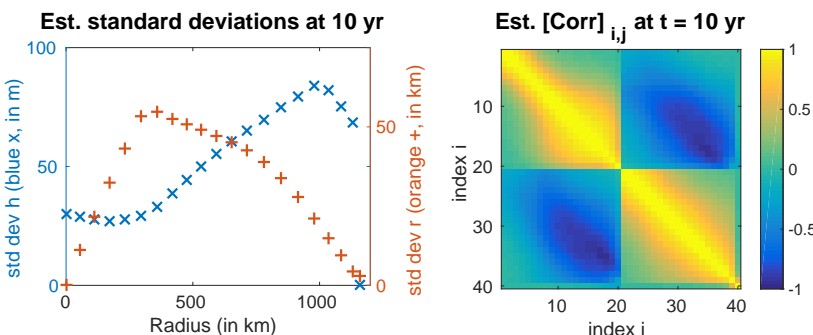

**Figure 12.** Standard deviations and correlation matrix $\mathbf{Corr}$ estimated from the analysis ensemble at time $t = 10$ yr in the advanced configuration where we observe surface elevation. The matrix $\mathbf{Corr}$ has the same structure as $\mathbf{B}$ defined by Eq. (27). Both standard deviations and correlation structures are similar to Fig. 7.

We now compare the ETKF with results obtained with 3D-Var. Absolute errors in the ice thickness at $r = 0$ and in the position of the margins are displayed for both cases in Fig 13. As in previous experiments, the ETKF performs better than 3D-Var. For example, the absolute error for the ice thickness at the ice divide stays below 60 m after $t = 1$ yr for the ETKF. By contrast, the absolute error for 3D-Var can be up to 125 m. The same statement remains valid for the absolute error in the position of the margin, which stays below 8 km for the ETKF after $t = 2$ yr, yet can be up to 20 km for 3D-Var. We remark

that, since the background state is smaller than the reference state, 3D-Var does not assimilate all available data. Indeed the algorithm cannot incorporate observations outside the background domain because of the form of the observation operator (see Eq. (31)). This is not, however, the case for the ETKF, even if the ensemble mean has a smaller domain than the reference




domain, since there is at least one member of the ensemble with a bigger domain than that of the reference. At the end both approaches show a similar accuracy in the forecast state after time $t = 10$ yr, showing again the efficiency of both DA schemes.

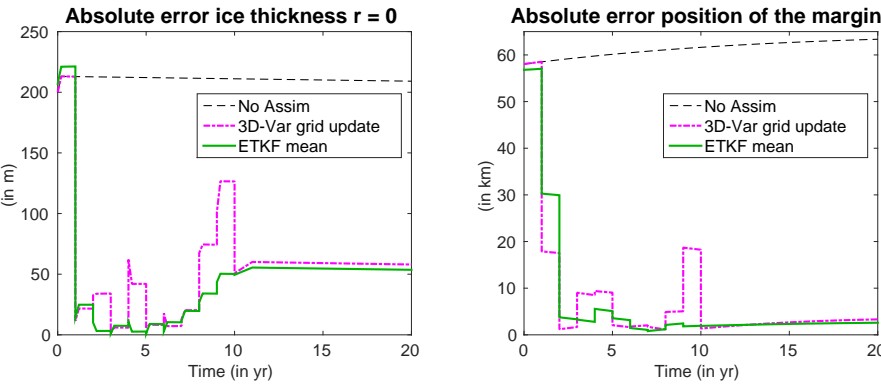

**Figure 13.** Evolution of the absolute error of the estimated ice thickness at $r = 0$ and the estimated position of the margin in the advanced configuration where we assimilate surface elevations over the first 10 years. We compare the absolute errors obtained when we use 3D-Var with the correction of the position of grid nodes and when we use an ETKF. The ETKF performs better than the 3D-Var for both variables.

## 5.3 Assimilating observations of surface velocity and position of the margin

We now consider assimilating observations of surface ice velocity and the position of the margin (if we only assimilate observations of surface ice velocity, the problem is undetermined).

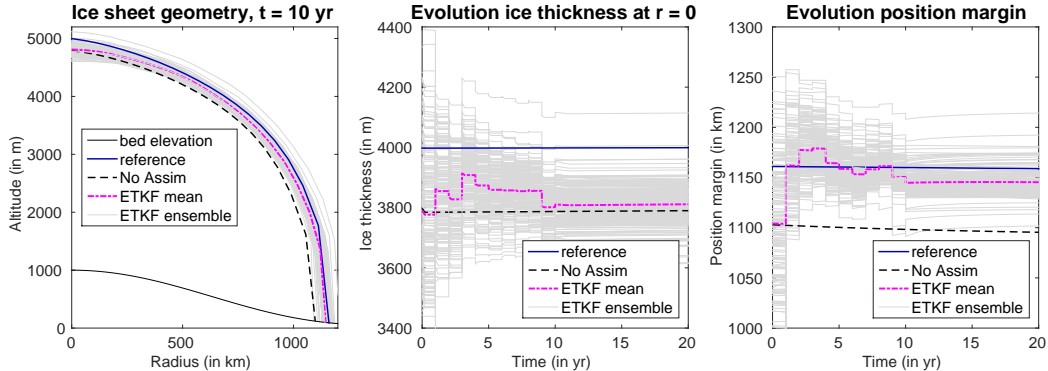

**Figure 14.** ETKF results for the advanced configuration where observations of surface ice velocity and the position of the margin are assimilated over the first 10 years and a forecast is made for the following 10 years. *Left:* ETKF analysis at time $t = 10$ yr compared with the reference. *Centre:* Evolution of the ice thickness at $r = 0$ with time. *Right:* Evolution of the position of the margin with time.

Again we want to compare the accuracy of 3D-Var and the ETKF using this new set of observations. We use the same background state, the same structure for **B** and the same initial ensemble as before. The observation operator for surface




velocities is nonlinear (see Eq. (34)). For that reason, even if the ensemble is large, inflation is mandatory for the ETKF. We take an inflation of $\lambda_{\text{infla}} = 1.10$. If the inflation is taken any larger in this example, the ETKF analysis produces ensemble members with a non-ordered grid and the experiment cannot be pursued.

We first study the results obtained with the ETKF. At the end of the DA window, $t = 10$ yr, the ice thickness profile is
retrieved well everywhere by the mean of the ensemble, except near the ice divide $r = 0$ (see Fig. 14). This is due to the relatively large uncertainty of surface velocity observations near the ice divide compared to the reference value at the same point (here $\sigma^o_{u_s} = 30\,\mathrm{m.yr}^{-1}$ and the reference surface velocity near the ice divide is below $0.1\,\mathrm{m.yr}^{-1}$). The estimated (mean) position of the margin at $t = 10$ yr, is 1144.7 km with an ensemble standard deviation of 12.1 km. This is an absolute error of 15 km, so is worse than in the case where we observed the surface elevation, but assimilating these data still provides better
estimates than those obtained with no assimilation. This comment remains valid for the forecasts obtained after $t = 10$ yr since estimates of the position of the margin are not degraded over the time window $[10\,\mathrm{yr},\,20\,\mathrm{yr}]$.

Estimates of the standard deviations and covariances, as shown in Fig. 15, differ from those of the previous experiment (see Fig. 12 for comparison). We observe that the standard deviations for the node positions are smaller in the middle of the ice sheet than in the previous experiment, but near the margin these are larger. The reduction in the standard deviation for ice
thickness variables close to the ice divide is less significant than in the previous experiment. This is due to the relatively large uncertainty of surface velocity observations near the ice divide compared to the reference value at the same point. We remark that assimilating observations of surface ice velocity together with the position of the margin leads to an increased correlation length scale for ice thickness variables and to a smaller correlation length scale for node positions compared to the previous experiment. Finally the cross-covariances have smaller anti-correlations and positive correlations appear between ice thickness
variables in the interior of the ice sheet and between node positions close to the margin. These differ significantly from the case where we assimilate observations of surface elevation as a result of the difference in observation operators.

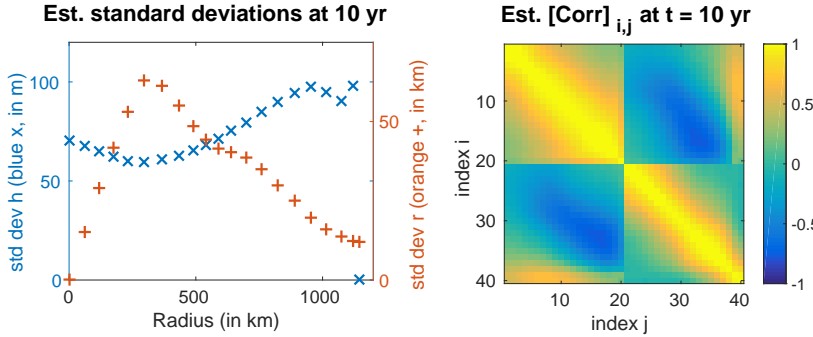

**Figure 15.** Standard deviations and correlation matrix **Corr** estimated from the analysis ensemble at time $t = 10$ yr in the advanced configuration where we observe surface ice velocity and the position of the margin. The matrix **Corr** has the same structure as **B** defined by Eq. (27). Both standard deviations and cross-correlation structures are different from those shown in Fig. 12.





We finally compare the ETKF with results obtained with 3D-Var. Absolute errors in the ice thickness at $r = 0$ and in the position of the margins are displayed for both cases in Fig 16. As in previous experiments the ETKF performs better than 3D-Var for the position of the margin. Nevertheless, 3D-Var still performs reasonably well in this nonlinear context. The forecast trajectory of the margin after $t = 10$ a is improved by DA in both cases. This demonstrates again the robustness of our DA

approach in the context of an ice sheet modelled with a moving point numerical model.

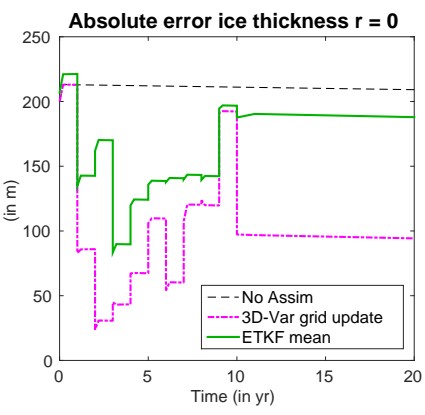
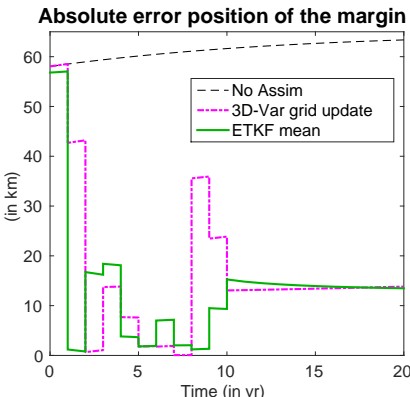

**Figure 16.** Evolution of the absolute error of the estimated ice thickness at $r = 0$ and the estimated position of the margin when we observe surface ice velocities and the position of the margin in the advanced configuration. We compare the absolute errors obtained when we use 3D-Var with the correction of the grid-node positions and when we use an ETKF. The ETKF performs better than the 3D-Var with respect to the position of the margin, but 3D-Var seems to give better results for the ice thickness at $r = 0$.

## 6 Conclusion and Prospects

In this paper we have adapted standard data assimilation techniques (a 3D-Var scheme and an ETKF) to estimate the state of a 1-d ice sheet model using a moving point method. This is done by including both ice thickness variables and the location of mesh nodes in the state vector. The only requirement is to ensure that the update does not produce a non-ordered moving mesh.

This can be achieved either by using an appropriate flow-dependent background covariance matrix with large correlations between adjacent mesh points or by using an ensemble with the same properties. This combination has been validated with various twin experiments assimilating classical available observations for an ice sheet (ice thickness, surface elevation and surface ice velocity) and also observations of the position of the boundary. These twin experiments show in particular that:

– the form of the state vector allows the explicit tracking of boundary positions for moving boundary problems;

– this form also allows a straightforward and efficient assimilation of boundary positions (in this paper, the position of the margin);

– assimilating spatially distributed observations gives better estimates if node locations are updated in the analysis step;



- – 3D-Var can have issues with assimilating observations if they are located outside the forecast domain; the ETKF can overcome these issues if at least one member of the ensemble has its numerical domain large enough to include the location of these observations;

- – the ETKF tends to provide better estimates than 3D-Var, mainly because of its capacity to remember past observations, but 3D-Var still provides satisfactory estimates;

- – ETKF provides not only good state estimates but also interesting information on the structure of the covariances; these are expected to be dominated by the statistics of the initial ensemble and the type of observations that are assimilated.

Whilst this paper uses a particular moving mesh method for the 1-d numerical model, our approach can be extended to any 1-d moving boundary problem modelled with a moving mesh, assuming only that the ordering of the points must be maintained. Moving mesh approaches are also suitable for modelling the evolution of 2-d moving boundary phenomena (Baines et al., 2009). The successful application of the moving mesh method to a 2-d model of an ice cap is presented in Partridge (2013). Initial results on the assimilation of observations of ice thickness into the 2-d ice cap model are also given in Partridge (2013). These results raise a number of issues concerning the approach needed for updating the nodal positions of the 2-d grid during the assimilation step. Research on these issues is on-going.

## Appendix A: Surface mass balances

### A1   EISMINT surface mass balance

For the twin experiments performed in Section 4 we use the simple constant-in-time surface mass balance employed in the moving margin experiments of the EISMINT intercomparison project (Huybrechts et al., 1996):

$$m(r) = \min\left(0.5\,\mathrm{m\,yr^{-1}}, 10^{-2}\,\mathrm{m\,yr^{-1}\,km^{-1}} \cdot (450\,\mathrm{km} - r)\right) \tag{A1}$$

### A2   Parametrised surface mass balance with feedback loop

For the twin experiments performed in Section 5 we use a more complex surface mass balance parameterised as a function of the surface atmospheric temperature $T_s(t,r)$. This simple parametrisation was used in Bonan et al. (2014) in the context of ice sheet model initialisation but with a fixed-grid model. The values of the different parameters involved in this parametrisation are given in Table 2. The surface mass balance is the sum of positive accumulation Acc (snow precipitation) and negative ablation Abl (melting) parametrised in Eq. (A2) and (A3).

$$\mathrm{Acc}(t,r) = \mathrm{Acc}_0\, e^{c_0\, T_s} \tag{A2}$$

$$\mathrm{Abl}(t,r) = \begin{cases} \mathrm{Abl}_0 \left(\dfrac{T_s - T_0}{T_0}\right)^2 & \text{if } T_s > T_0 \\ 0 & \text{otherwise} \end{cases} \tag{A3}$$





The surface temperature depends on the altitude of the surface $s$, the distance from the origin and a climate temperature $T_{\mathrm{clim}}(t)$ evolving in time according the relation

$$T_s(t,r) = T_{\mathrm{clim}}(t) + \lambda r + \gamma s(t,r) \tag{A4}$$

This parametrisation aims to reproduce qualitatively a typical surface mass balance over an ice sheet and to include feedbacks
5   associated with the evolution of the geometry.

**Table 2.** List of parameter values used for the parameterised surface mass balance

| Parameter | | Value |
|---|---|---|
| $\mathrm{Acc}_0$ | rate of accumulation | $6\ \mathrm{m.yr}^{-1}$ |
| $\mathrm{Abl}_0$ | rate of ablation | $-5\ \mathrm{m.yr}^{-1}$ |
| $T_0$ | minimum temperature for ablation | $-6\ ^{\circ}\mathrm{C}$ |
| $c_0$ | coefficient exponential law for accumulation | $0.115\ ^{\circ}\mathrm{C}^{-1}$ |
| $\lambda$ | longitudinal gradient of surface temperature | $\dfrac{1}{111000}\ ^{\circ}\mathrm{C.m}^{-1}$ |
| $\gamma$ | vertical gradient of surface temperature | $-0.0063\ ^{\circ}\mathrm{C.m}^{-1}$ |

*Acknowledgements.* This research was funded in part by the Natural Environmental Research Council National Centre for Earth Observation (NCEO) and the European Space Agency (ESA).

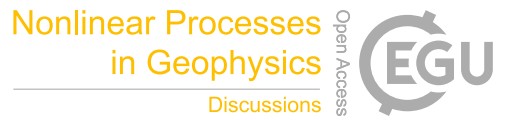

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
