# Peer review of "Data assimilation for moving mesh methods with an application to ice sheet modelling"

_Nonlinear Processes in Geophysics, 2016_

## Referee Comment (RC1) · Anonymous Referee #1 · 12 Oct 2016

The paper presents experiments on assimilation of observations of the flow of an ice sheet. A specific, and original, feature of the paper is that, following an earlier work of the authors (Bonan *et al.*, 2016), the flow model is built on a moving mesh that allows description of the motion of the front of the ice sheet while avoiding unpleasant spatial interpolations. The experiments are of the identical twin type, in which the 'observations' that are assimilated are extracted from an earlier integration of the assimilating model. The authors use two different assimilation algorithms, one that they call '3D-Var' (see comment 12 below about the use of that expression), and a standard Ensemble Transform Kalman Filter (ETKF). Both algorithms are basically successful, with a general advantage for the latter.

The paper is well written and instructive, and I think its material is worth publishing, provided however a number of improvements are made. I give below my scientific comments and suggestions (in approximate order of decreasing importance), followed by editing remarks.

**Scientific comments**

1. There seems to be a basic contradiction in what the authors exactly do. It is said in subsection 2.2 (*Moving-point method*) that the evolution of the node position $^\wedge r$ is determined by the condition that the corresponding mass fraction $\mu(^\wedge r)$ is conserved in time (Eq. 9, later discretized in the form 11). It is then said in the following subsection that the discretized mass fractions $\mu_i$ are updated in the temporal integration (l. 20, p. 8, and Eq. 15).

The authors also write (eq. 26) that the state vector **x** of their model consists of the ice thicknesses $h_i$ at the points of the moving nodes, and of positions $^\wedge r_i$ of those nodes There should then be, in agreement with the general equation (16), prognostic equations for both the $h_i$'s and $^\wedge r_i$'s, There is one for $^\wedge r_i$'s, but none for the $h_i$'s. On the contrary, $h$ is defined diagnostically by eq. (12) from the profile of the mass fraction.

I consider it is necessary for acceptance of the paper to first resolve that contradiction. I suspect that what is described in subsection 2.2 is actually not what is done in the numerical model. What is done must be precisely described. In particular, the prognostic equation for the thickness $h$ (if there is any) must be explicitly mentioned. And, if the mass fractions are not conserved, by what is the evolution of the node positions $^\wedge r_i$ determined ?

2. The authors find that ETKF performs generally better than '3D-Var' (one exception is actually shown on the left panel of Fig. 16; and, on Fig.13, the performance of both algorithms is the same at the end of the 10 years of assimilation). They give as an explanation of the better performance of ETKF the fact that the latter 'remembers' past observations (p. 17, l. 1, p. 24, l. 4). That really does not explain much. Actually '3D-Var' also 'remembers' past observations as seen in many places in the paper (*e.g.*, right panel of Fig. 16) where it produces forecasts that are much more accurate than what is obtained when no observations are used.

It is possible to say more. The basic difference between '3D-Var' and ETKF is that the former uses a background error covariance matrix **B** that is defined from the start and remains static in time, while the latter computes a matrix **B** from an ensemble of forecasts. Both algorithms evolve in time an estimate of the state of the observed system, but ETKF evolves in addition an estimate of the associated estimation error. That very likely leads to a better estimate of the matrix **B** and to a better analysis. Actually, it is very generally observed that assimilation algorithms that carry in time an estimate of the estimation error (such as 4D-Var and the various forms of Kalman Filter) perform better than algorithms that use a static estimate of the estimation error (as does 3D-Var).

The importance of evolving in time an estimate of the estimation error must be stressed.

And I suggest the authors look in more detail at the consistency of the 'predicted' and 'observed' background. The authors state repeatedly that the reference lies in their ETKF experiments

within the range of the predicted ensemble. What about the '3D-Var' ? It does not produce an explicit range, but it uses a known matrix **B** to which the observed background error can be compared. And how does the matrix **B** compare between the two algorithms ? Does it tend to be larger in either one of them? Such a comparison can fundamentally be made from Figs 3 and 7, but these figures are not in the same format (covariances in Fig. 3, correlations in Fig. 7).

3. An important question in assimilation is the degree of stability of the observed system. That question is always present, but is particularly obvious in sequential assimilation, such as 3D-Var and ETKF. In sequential assimilation, the evolution of analysed state $x^a$ results of the combined influence of, on the one hand, the unstable modes of the system and of the possible model errors, which together tend to increase the estimation error, and on the other hand, of the stable modes and the introduction of observations at analysis time, which together tend to decrease the estimation error (there are also neutral modes, which have at most marginal effect on the evolution of the uncertainty). Were there instabilities in the present case ? The fact that the forecast error tends to remain constant after the end of the assimilations (see, e. g., Fig. 16) suggests that there are no instabilities in the system. That is not surprising for the motion of highly viscous fluid. But assimilation, in the case of a system which has no instabilities used with an error-free model (which is the case here) is relatively easy. Any assimilation algorithm, unless it is devised or implemented in a particularly unfortunate way, will normally be 'successful' in that it will gradually move the analyzed state towards the reference from which observations are extracted.

This seems to be the case here. That does not decrease the value of the paper for the study of ice sheet modelling. But it should be mentioned that the problem considered in the paper is relatively easy from the point of view of the assimilation, in that all difficulties that might result from the presence of instabilities in the system and/or of model errors are absent.

4. A number of statements are made (for some as a way of explanation of a specific observed feature) which seem unjustified.

- 4.1. P. 12, ll. 5-6, … *this approach ensures that this moving-point framework produces positive estimates of ice thickness variables and a smooth interior profile …* What is the evidence, particularly as concerns the positiveness of thickness variables (I presume the question can arise only in the vicinity of the ice margin) ?

- 4.2. P. 13, ll. 15-16, *The formulation of* **B**$_r$ *aims to ensure that the order between mesh points defined by Eq. (13) is preserved by the 3D-Var algorithm. Since the distance between nodes evolves in time, it is even more important than in the previous case to use a flow-dependent background error covariance matrix* **B**. In what does the particular formulation of **B**$_r$ helps preserving the order of mesh points ?

- 4.3. P. 16, ll. 17-18. …*the experiment shows the sensitivity of 3D-Var to current observations because of the use of fixed variances in the prescribed covariance matrix* **B**. I am not sure to understand what you mean. But if you imply that the increase of error at the second analysis time is due to the use of fixed variances, that is unfounded.

- 4.4. P. 20, l. 4, *This is because the ensemble spread is too small in that area.* The middle panel of Fig. 11 does not show a smaller ensemble spread beyond year 7.

- 4.5. P. 21, l. 10, and p. 22, l. 1, *The observation operator for surface velocities is nonlinear* (…). *For that reason, even if the ensemble is large, inflation is mandatory for the ETKF.* Is there any objective evidence for a link between the nonlinearity of the observation operator and the need for

inflation of the analyzed ensembles ? I suggest you only mention that you have observed that, even though the ensembles are large, inflation is necessary in the present case.

5. The authors write in the conclusion (p. 24, ll. 1-3) *3D-Var can have issues with assimilating observations if they are located outside the forecast domain; the ETKF can overcome these issues if at least one member of the ensemble has its numerical domain large enough to include the location of these observations*. The problem raised by observations that are located outside the background domain is hardly discussed in the main text (from p. 20, l. 19 to p. 21, l. 1). If they consider that aspect to be important enough to be mentioned in the conclusion, the authors must discuss it at more length in the main text, and not wait the presentation of the 'advanced configuration' experiments to discuss it.

I first mention that what I understand of the first line of p. 21 would be better expressed as … *since it is observed that there is always at least one member of the ensemble whose domain is larger than the domain of the background* (I think that, contrary to what the authors write, it is the background that matters here, not the reference).

Second, what is done in 3D-Var with those outlying observations ? Are they simply ignored ?

And, in ETKF, even if one or more ensemble members can accommodate those observations, but not all, how is the ensemble increased back after the analysis to its full dimension $N_e$ ?

6. Eq. (34-35). I understand that equation originates from Eq. (4). Say it. And give explanations (or at least a reference) for the complicated expression on the right-hand side.

7. Subsection 5.3. The errors on the observations of surface velocity and of position of the margin are apparently not mentioned.

8. P. 23, caption of Fig. 16, last two lines. You write *The ETKF **performs better** than the 3D-Var with respect to the position of the margin, but 3D-Var **seems** to give better results for the ice thickness at r = 0.* Do you imply the better performance of ETKF on the left panel of the figure is real, but the better performance of 3D-Var on the right panel might be only apparent ?

9. Figures 4 and 5. The left panel shows results at the first analysis time $t_1$ = 500 yr. It would be preferable to show also the analogous results at the second time $t_2$ = 1500 yr.

10. Since the surface altitude $s(t, r)$ is known, there is no need for making a distinction between ice thickness and surface elevation (Eqs 30 and 31). These observations are exactly equivalent.

**Editing comments**

11. Literally speaking, the expression *3D-Var* is inappropriate for the paper. The physical problem under consideration is one-, not three-dimensional, and the authors mention nowhere that they have used an explicit variational algorithm for determining the analysed state $\mathbf{x}^a$. Now, their algorithm possesses a property which, owing to a pure language convention in the trade of assimilation, is associated with the expression 3D-Var. Namely, that is uses a time-invariant background error covariance matrix $\mathbf{B}$ (that convention originates from ECMWF, which developed a 3D-Var as a preliminary step towards its full-fledged 4D-Var). I suggest that the authors, if they want to keep the expression 3D-Var, say clearly that their algorithm is neither 3D nor variational, and use a different notation ('*3D-Var*', *3D-Var-like, …*)

12. The right-hand side of Eq. (21) should read $\mathbf{B}_k \mathbf{H}_k^{\mathrm{T}}(…)$ (and not $\mathbf{B}_k \mathbf{H}_k$)

13. P. 18, l.10. You mention $T_{clim}$. Reference should be made here to Eq. (A4).

14. Eq. (A3). I think the unit for $T$ is kelvin. Say it.

---

## Referee Comment (RC2) · S. L. Cornford (Referee) · 28 Oct 2016

General Comments ——————————-

Bonan et al discuss a data assimilation (DA) technique applied to a moving mesh ice sheet model. The promise lies in its ability to add observations of ice sheet margin position, and indeed, correctly account for the motion of that margin.

The paper employs a rather simplified 1D description of ice sheet physics, so I would tend to see it as sketch of a technique that might be useful in the more complex 2 or 3 D problems currently of interest. Given that, I would hope to be able to assess the value in developing such methods further, but the problem studied is just too far away from the problems of interest for me to feel any the wiser.

[Figure]

The ice sheet physics is the 'shallow ice approximation', which I think it is fair to say is of little interest in contemporary ice dynamics. It is still of (diminishing) interest in the study of the distant past (ie the rise and fall of ice ages), and it is possible to imagine this sort of technique being of great interest there *if* it could be used in the right kind of data assimilation. The data would be sparse in both space and time - isolated values ( ~1 point in the whole domain) for past surface elevations of ice sheets where their surface intersected with rock, and some observations of their margins/extent through time from depositions, landscape scouring and so on. The synthetic data in this paper is very much more like contemporary satellite data – dense observations of surface velocity – only available at all through satellite observation – and elevation.

Given that the paper is dealing with contemporary ice sheet change, the shallow ice approximation is not sufficient. It neglects both membrane (xx,yx,yy) stress and vertical shear (xz,yz) stress at the bed, and considers only vertical (xz,yz) shear within the ice. These neglected processes are thought to be involved in every case of contemporary dynamical change, whether that is surging glaciers (a change in sliding) or the loss of buttressing mechanisms (a change in boundary condition, in the simplest kind of treatment), whereas in-ice zx-stresses are seen as unimportant.

It is possible that the shallow ice approximation (eq 4) is mathematically close enough to the systems of interest to imagine the DA methods being of wider value. I'm simply not sure. In 1D eq 4 would be replaced by a nonlinear elliptic equation in U(x) and in the most simple case a boundary condition on U(rl) from Schoof 2007 - which incidentally implies a non-zero flux across the margin (the grounding line) so that eq 7 for the mesh movement is not right. It is true that the elliptic equation can be approximated by something like eq 4 far from the margin, but that is invalid in the fast sliding glaciers that are seeing present day change.

Does that fact that eq 4 should be an elliptic PDE, rather than a simple expression, matter to the conclusions of the paper? It might. It means that eq 34 also requires the solution to an elliptic PDE, so that in turn it is harder to find dH/dx_h and dH/dx_r in

eq 27. That is possible though, indeed, most recent progress in ice sheet DA involves such calculations. It also must have some impact on the ETKF method, because each member is much more expensive - how well will ETKF perform if the number of samples is limited? The 200 members used here might be the practical limit in a 2D or 3D problem, but there would be more degrees of freedom. I would have been interested to see how well/poorly ETKF performed with, say, 20 members

Overall, I'd say that unless the general mathematics of the moving mesh and DA approach are ineresting in themselves (hard for me to judge, but they seem to be relatively straightforward), then this work needs to be based on more relevant ice sheet physics - I would suggest the models of Schoof 2007 are the right place to start - or the reader needs to be convinced that the eq 4 is adequate for reasons beyond the physics.

---

## Referee Comment (RC3) · S. L. Cornford (Referee) · 4 Nov 2016

The authors suggest "Given the comments of the reviewer, we would plan to augment the paper with some examples using sparse data and others using fewer ensemble members to demonstrate the power and limitations of this approach"

I think that should be sufficient to address my concerns

---

## Author Comment (AC1) · 4 Nov 2016

Response to Reviewer 2:

The research addresses the use of data assimilation with new numerical techniques for modelling moving boundary problems. We illustrate our approach on ice flow with the aim of efficiently obtaining more accurate estimates for the margins of ice sheets. A relatively simple model of ice flow is used here to investigate the new techniques. The new moving-mesh numerical methods for the ice flow have already been validated for both 1-D and 2-D models of ice flow (see [1] and [2]). The aim of this paper is to demonstrate that it is possible to combine sophisticated data assimilation methods with a moving mesh numerical modelling technique. Given that the techniques are successful on the simplified problem, there is no reason that these cannot be extended

to much more complex problems. The major advantage of the moving mesh method is that a only small number of mesh steps is needed to accurately determine the boundary positions of the flow, unlike adaptive mesh methods.

In response to the comments:

The method works effectively even with sparse observations, see [2], but the scenario with more dense satellite data is much more realistic now.

The fact that an elliptic problem needs to be solved at each step of the model is the same for any other model of the flow. Here we use a direct solution in the simplified case, but this is not necessary. The elliptic problems can be solved in parallel numerically at each time step in the ensemble filter assimilation method, but if computational power is a constraint, then the 3DVar method gives good results without solving multiple elliptic problems.

Given the comments of the reviewer, we would plan to augment the paper with some examples using sparse data and others using fewer ensemble members to demonstrate the power and limitations of this approach.

Bertrand Bonan, Nancy Nichols, Mike Baines and Dale Partridge

[1] Bonan B. et al., The Cryosphere, 10, 1-14 2016

[2] Partridge, PhD Thesis, University of Reading, 2013

---

## Editor Comment (EC1) · O. Talagrand (Editor) · 9 Nov 2016

Two referees have now sent their reports on the paper. Referee 1, who is a specialist of assimilation of observations, makes comments on the methodological aspects of the paper. His main concern is about what he considers to be a contradiction between the mathematical presentation of the ice flow model given in subsection 2.2 of the paper and what is apparently done in the actual numerical implementation of the model.

Referee 2, who is a specialist of ice flow modelling (and has let his name known) is more critical in that he considers the experimental conditions of the work presented in the paper are too simple to be really instructive, and that it is not clear whether the method used by the authors would work satisfactorily in more realistic conditions. The authors have already responded the referee's comments, by saying that they plan to perform additional experiments using sparser data and fewer ensemble members for the ETKF algorithm. The referee has immediately responded that this would be sufficient to address his concerns.

My suggestion as Editor is therefore that the authors do what I presume they have already started doing, namely prepare a new version of the paper including the results of the additional experiments they have announced. The new version must be prepared according to the instructions the authors have received from the Editorial Office of *Nonlinear Processes in Geophysics*. In particular, the authors must give a point-by-point response to all of both referees' comments and requests. Should they disagree with one particular comment, or decide not to follow one particular request, they must state precisely their reasons for that.

If, as I hope, the authors submit a new version of their paper, that version will be submitted to further review by (normally) two referees who may, or may not, be the same as those of the first version.

---

## Author Response (AR1)

We thank the referees for their comments and advice and for their positive remarks . We have made amendments to the paper in response to the comments, which we detail below. The reviewer's comments are shown in blue and the responses in black. Changes made to the paper are highlighted here and in the text.

**Response to Reviewer 1:**

1. There seems to be a basic contradiction in what the authors exactly do. It is said in subsection 2.2 (*Moving-point method*) that the evolution of the node position $^\wedge r$ is determined by the condition that the corresponding mass fraction $\mu(^\wedge r)$ is conserved in time (Eq. 9, later discretized in the form 11). It is then said in the following subsection that the discretized mass fractions $\mu i$ are updated in the temporal integration (l. 20, p. 8, and Eq. 15).

During the integration of the dynamic equations, the mass fractions are not updated. These are keep constant in accord with the equation $d\mu(r)/dt = 0$. We have added this equation for clarity in the text after equation (8). At the *initial* time, the mass fractions are determined from the specified ice thickness and mesh points using equations (14) - (15). These are not updated during the evolution of the system.

The authors also write (eq. 26) that the state vector **x** of their model consists of the ice thicknesses $h i$ at the points of the moving nodes, and of positions $^\wedge ri$ of those nodes There should then be, in agreement with the general equation (16), prognostic equations for both the $h i$'s and $^\wedge ri$'s, There is one for $^\wedge ri$'s, but none for the $h i$'s. On the contrary, $h$ is defined diagnostically by eq. (12) from the profile of the mass fraction. I consider it is necessary for acceptance of the paper to first resolve that contradiction. I suspect that what is described in subsection 2.2 is actually not what is done in the numerical model. What is done must be precisely described. In particular, the prognostic equation for the thickness $h$ (if there is any) must be explicitly mentioned. And, if the mass fractions are not conserved, by what is the evolution of the node positions $^\wedge ri$ determined ?

There is a misunderstanding here. The numerical model evolves both the states of the system (ice thickness) and the mesh points simultaneously forward in time in such a way as to maintain constant mass fractions as defined at the initial time by (14)-(15). The mass fractions are not updated during the evolution of the system. The prognostic equation is implicit and given by $d\mu(r)/dt = 0$, as explained above. The mesh points are evolved using a discrete form of equation (9) and the ice thickness is determined using a discrete form of (12) at each time-step. Full details are given in the open access paper: B. Bonan, M. J. Baines, N. K. Nichols, and D. Partridge. A moving-point approach to model shallow ice sheets: a study case with radially symmetrical ice sheets. The Cryosphere, 10:1–14, 2016. doi: 10.5194/tc-10-1-2016.

The numerical method is precisely as described here, but we tried to keep the description of the method to a minimum, since the details are given elsewhere and the objective of the paper is not to derive the numerical method here. We have, however, now added a sentence to the end of Section 2.3 to make the method more explicit. We have added (p.5 lines 17-18): 'The mesh points are then evolved using a discrete form of equation (9) and the ice thickness is determined using a discrete form of (12), with the mass fractions $\{\mu_i\}$ kept constant over a time step. Full details are given in (Bonan et al, 2016).'

We remark that, at an assimilation step, the states and mesh points are all updated using either the ensemble method or the 3D-Var (best linear) estimate, as described in Section 3. At this point the mass fractions then need to be recalculated (updated) to take into account the change in the ice thickness and new positions of the mesh points. This is done with a numerical approximation to the integral (8) as in (14) - (15) - ie the model is reinitialized as described in section 3.2. The states and mesh points are then again evolved forward to the next assimilation time keeping the mass fractions constant.

2. The authors find that ETKF performs generally better than '3D-Var' (one exception is actually shown on the left panel of Fig. 16; and, on Fig.13, the performance of both algorithms is the same at the end of the 10 years of assimilation). They give as an explanation of the better performance of ETKF the fact that the latter 'remembers' past observations (p. 17, l. 1, p. 24, l. 4). That really does not explain much. Actually '3D-Var' also 'remembers' past observations as seen in many places in the paper (*e.g.*, right panel of Fig. 16) where it produces forecasts that are much more accurate than what is obtained when no observations are used. It is possible to say more. The basic difference between '3D-Var' and ETKF is that the former uses a background error covariance matrix **B** that is defined from the start and remains static in time, while the latter computes a matrix **B** from an ensemble of forecasts. Both algorithms evolve in time an estimate of the state of the observed system, but ETKF evolves in addition an estimate of the associated estimation error. That very likely leads to a better estimate of the matrix **B** and to a better analysis. Actually, it is very generally observed that assimilation algorithms that carry in time an estimate of the estimation error (such as 4D-Var and the various forms of Kalman Filter) perform better than algorithms that use a static estimate of the estimation error (as does 3D-Var). The importance of evolving in time an estimate of the estimation error must be stressed. And I suggest the authors look in more detail at the consistency of the 'predicted' and 'observed' background. The authors state repeatedly that the reference lies in their ETKF experiments within the range of the predicted ensemble. What about the '3D-Var'? It does not produce an explicit range, but it uses a known matrix **B** to which the observed background error can be compared. And how does the matrix **B** compare between the two algorithms? Does it tend to be larger in either one of them? Such a comparison can fundamentally be made from Figs 3 and 7, but these figures are not in the same format (covariances in Fig. 3, correlations in Fig. 7).

The statement that the ETKF 'remembers' past observations, was intended to convey the point made by the referee here, that the ETKF evolves the background error covariance matrix using the observations and hence 'remembers' information obtained from these, which is not achieved by the '3D-Var' approach, where the background error covariance has a static structure. Hence the ETKF provides flow-dependent background error covariances which generally give better results than '3D-Var'. We have made these statements more explicit and clarified this point, adding a figure and further discussion on p.18. See the response to point 4.3 below for full details. On p.18 (line 14) we have replaced the sentence 'Remembering past observations ensures that……' by the sentence 'Propagating the background error covariances using the ensemble statistics ensures that…' and on p.25 (lines 11-12) we have replaced the statement 'to remember past observations' with the statement 'to provide flow dependent statistical estimates of the background error covariances'.

The comparison between the covariances obtained by the ETKF and those that are used in the '3D-Var' method is interesting. The main point to be observed is the cross correlations between states and mesh points that are produced by the ETKF in comparison to those specified in the '3-DVar' method. The question of whether the reference lies within the assumed variability for the 3DVar case is also interesting. We have added a further figure, Figure 6, to Section 4.3 for the '3-DVar' case, which is similar to the Figure 8 for the ETKF case, to show the standard deviations and the estimated analysis covariance matrix obtained in the 3DVar case. We have also added discussion of the comparison between the results for the '3-DVar' and ETKF cases to Section 4.5. See the response to point 4.3 below.

3. An important question in assimilation is the degree of stability of the observed system. That question is always present, but is particularly obvious in sequential assimilation, such as 3D-Var and ETKF. In sequential assimilation, the evolution of analysed state **x**a results of the combined influence of, on the one hand, the unstable modes of the system and of the possible model errors, which together tend to increase the estimation error, and on the other hand, of the stable modes and the introduction of observations at analysis time, which together tend to decrease the estimation error (there are also neutral modes, which have at most marginal effect on the evolution of the uncertainty). Were there instabilitiesin the present case ? The fact that the forecast error tends to remain constant after the end of the assimilations (see, e.g., Fig. 16) suggests that there are no instabilities in the system. That is not

surprising for the motion of highly viscous fluid. But assimilation, in the case of a system which has no instabilities used with an error-free model (which is the case here) is relatively easy. Any assimilation algorithm, unless it is devised or implemented in a particularly unfortunate way, will normally be 'successful' in that it will gradually move the analyzed state towards the reference from which observations are extracted. This seems to be the case here. That does not decrease the value of the paper for the study of ice sheet modelling. But it should be mentioned that the problem considered in the paper is relatively easy from the point of view of the assimilation, in that all difficulties that might result from the presence of instabilities in the system and/or of model errors are absent.

We agree that the ice sheet model is essentially diffusive, but it is highly nonlinear and we might expect some difficulties to arise with an assimilation method based on linear and Gaussian assumptions. This does not turn out to be a problem here since both methods perform quite well. The primary objective of the paper is to show that assimilation in the moving mesh method is feasible and can easily and efficiently be implemented. Traditionally numerical techniques for tracking fronts accurately use adaptive mesh methods in which assimilation methods are very expensive and difficult to implement. We have added more on this point in the introduction, which has been rearranged in order to emphasize the objective of the paper, and have provided further references for papers on adaptive methods and applications to ice sheet modelling (see p. 1 line 18 in particular ).

We note that filter methods act as a feedback mechanism and, even in the case where the observed system is unstable, the filter acts as a stabilizing mechanism that in general ensures that the closed loop DA system is itself stable. That is the result here as is seen in the experiments. A mathematical proof is beyond the scope of this paper.

4. Comments:   A number of statements are made (for some as a way of explanation of a specific observed feature) which seem unjustified.

4.1. - P. 12, ll. 5-6, … *this approach ensures that this moving-point framework produces positive estimates of ice thickness variables and a smooth interior profile* … What is the evidence, particularly as concerns the positiveness of thickness variables (I presume the question can arise only in the vicinity of the ice margin) ?
We have reworded this sentence to read (p.12-13 lines 18-2):   'Nevertheless, our experiments demonstrate that this formulation of the background error covariance matrix ensures that the moving point framework produces positive…'

4.2. - P. 13, ll. 15-16, *The formulation of* **B***r aims to ensure that the order between mesh points defined by Eq. (13) is preserved by the 3D-Var algorithm. Since the distance between nodes evolves in time, it is even more important than in the previous case to use a flow-dependent background error covariance matrix* **B**.  In what does the particular formulation of **B***r helps preserving the order of mesh points ?
We have modified this sentence to read (p.13-14  lines 18-1):   'The correlation matrix B_r constrains the movement of the assimilated mesh points and the correlation function used in the formulation of B_r   is selected to ensure that the order of the points defined by Equation (13) is preserved.'

4.3. - P. 16, ll. 17-18. ... *the experiment shows the sensitivity of 3D-Var to current observations because of the use of fixed variances in the prescribed covariance matrix* **B**. I am not sure to understand what you mean. But if you imply that the increase of error at the second analysis time is due to the use of fixed variances, that is unfounded.
We believe that because of the fixed structure of the covariances, which now depend on the background positions of the mesh points, the updated covariance matrices are not sufficiently accurate in this case and cause a loss of accuracy in the filter. We have rephrased this sentence as follows (p. 18 lines 2-3):  '…... the experiment shows the sensitivity of 3D-Var to current observations resulting from the dependence of the prescribed covariance matrix **B** on the positions of the mesh nodes.'

We have also added a figure (Figure 6, p. 15) to Section 4.3 showing the analysis error variances and correlations obtained by the '3-DVar' procedure, which in comparison to the corresponding figure for the ETKF procedure (Figure 8) in Section 4.4, supports this conclusion. We have added the text (p.14 lines 11-17): 'The 3DVar method provides information on the analysis covariance structures for ice thickness variables and mesh point positions. In Fig. 6 we display the estimated standard deviations and the error correlation matrix Corr (see Eq. (26)) obtained at time t = 500 yr. using the estimated analysis error covariance matrix $P_{e,k}$ given by Eq. (22). We see that the 3DVar method produces decreased standard deviations and correlation length scales for ice thickness variables close to the ice divide and decreased standard deviations and correlation length scales for node locations close to the margin. The 3DVar method also produces strong anti-correlations between ice thickness variables and node positions, meaning that in order to fit the observations where the ice thickness variables become larger, the associated nodes need to retreat.'

In Section 4.5 we have added a further figure (Figure 10, p. 18) showing the evolved background covariance matrices used at time $t2$ = 1500 yr to obtain the analysis and have added the following text (p.18 lines 6-16): 'This improvement can be attributed to the better background forecast produced by the ETKF at each assimilation time.
In Figure 10 we display the background error covariance matrices used by the 3DVar and ETKF methods to produce the analysis at time t = 1500 yr. At the previous assimilation time t = 500 yr., the analysis covariances produced by both methods are very similar, as seen in Figs. 6 and 8. However, because the 3DVar error covariance matrix has a fixed form, the background covariance matrix used by 3DVar at the assimilation time t = 1500 yr. has not changed significantly. In contrast it can be seen that the ETKF background error covariance has fully evolved and contains much more information than the 3DVar error covariance matrix. This explains the better ability of the ETKF to provide accurate estimates in the context of the moving point model. Propagating the background error covariances using the ensemble statistics ensures that the ETKF is a more reliable scheme than 3D-Var. This improvement has a computational cost, however, as we now need to run the model $N_e$ times instead of once for 3D-Var.'

4.4 - P. 20, l. 4, *This is because the ensemble spread is too small in that area*. The middle panel of Fig. 11 does not show a smaller ensemble spread beyond year 7.
The ensemble spread has become too small at year 7 and the ensemble remains too small after that point. We have replaced the phrase ' is too small in that area' by the phrase (p.21 line 12): 'is too small from that time onwards'.

4.5. - P. 21, l. 10, and p. 22, l. 1, *The observation operator for surface velocities is nonlinear (…). For that reason, even if the ensemble is large, inflation is mandatory for the ETKF*. Is there any objective evidence for a link between the nonlinearity of the observation operator and the need for inflation of the analyzed ensembles ? I suggest you only mention that you have observed that, even though the ensembles are large, inflation is necessary in the present case.
We agree and have changed the sentence to read (p. 22 line 18): '…and even though the ensemble is large, inflation is necessary in this case.'

5. The authors write in the conclusion (p. 24, ll. 1-3) *3D-Var can have issues with assimilating observations if they are located outside the forecast domain; the ETKF can overcome these issues if at least one member of the ensemble has its numerical domain large enough to include the location of these observations*. The problem raised by observations that are located outside the background domain is hardly discussed in the main text (from p. 20, l. 19 to p. 21, l. 1). If they consider that aspect to be important enough to be mentioned in the conclusion, the authors must discuss it at more length in the main text, and not wait the presentation of the 'advanced configuration' experiments to discuss it.

We agree and have added a paragraph at the end of Section 3.2 pointing out this issue as follows (p. 9 lines 10-12): 'We remark that observations outside the domain of the background state at the

time of the update cannot be assimilated. This is a limitation on both methods, but the ETKF has the advantage that it can take into account such observations if the domain of the background of any member of the ensemble is large enough to include the reference domain.'

I first mention that what I understand of the first line of p. 21 would be better expressed as … *since it is observed that there is always at least one member of the ensemble whose domain is larger than the domain of the background* (I think that, contrary to what the authors write, it is the background that matters here, not the reference).

The observations are taken from the reference and hence all observations are within the domain of the reference; therefore we need one ensemble member with a domain larger than that of the reference to be able to incorporate all the observations. So the statement is correct as it is (p.22 lines 10-11) .

Second, what is done in 3D-Var with those outlying observations ? Are they simply ignored ?

As already stated (p.22 line 8), 'since the background state is smaller than the reference state, 3D-Var does not assimilate all available data.' - in other words, these observations are ignored.

And, in ETKF, even if one or more ensemble members can accommodate those observations, but not all, how is the ensemble increased back after the analysis to its full dimension *Ne* ?

All the ensembles are used at each assimilation step. Each ensemble is updated with the observations only within the domain of the background of that ensemble member. As with the 3DVar scheme, observations outside the background domain are ignored in updating each ensemble member.

6. Eq. (34-35). I understand that equation originates from Eq. (4). Say it. And give explanations (or at least a reference) for the complicated expression on the right-hand side.

The expression derives from Equation (4) together with a linear interpolation operator, as already indicated. We have added the expression (p.9 line 24) 'from a discretization of Eq. (4)' to the sentence.

7. Subsection 5.3. The errors on the observations of surface velocity and of position of the margin are apparently not mentioned.

This information was already defined in Subsection 5.1 (p.20 lines 11-16) before the results were discussed in Subsections 5.2 and 5.3; it is given as: 'We generate observations of surface elevation, surface ice velocity and the position of the ice sheet margin at times $t = 1,2,…,10$ yr from the reference run. The observations of the surface are taken at each point including the margin with an added Gaussian noise (uncorrelated with standard deviation $\sigma^o_s = 200$ m). The observations of the surface ice velocity are located at the mid-points between mesh points (so we have 20 observations of surface velocity). Observations are noised using a Gaussian law (standard deviation $\sigma^o_{us} = 30$m yr$^{-1}$, uncorrelated). For the position of the margin, the observational noise is sampled from N(0; $(\sigma^o_r{}^2)$ with $\sigma^o_r = 50$ km.'

8. P. 23, caption of Fig. 16, last two lines. You write *The ETKF **performs better** than the 3DVar with respect to the position of the margin, but 3D-Var **seems** to give better results for the ice thickness at r = 0.* Do you imply the better performance of ETKF on the left panel of the figure is real, but the better performance of 3D-Var on the right panel might be only apparent ?

We have rephrased this sentence (Figure 18, Caption, p.25) to read: '…but 3D-Var gives better results for the ice thickness at r = 0 in this case.'

9. Figures 4 and 5. The left panel shows results at the first analysis time $t1 = 500$ yr. It would be preferable to show also the analogous results at the second time $t2 = 1500$ yr.

In these figures the general behaviour of the solution at $t2$ is shown in the middle and right panels. But at time $t2$ the solutions are very close together and it is difficult to distinguish the significant behaviour in the assimilation schemes at this point. We do not feel that showing the results at $t2$ in addition to those at $t1$ would provide further information and therefore we have not illustrated the results at time $t2$. The important difference in behaviour between the results shown in these two figures occurs at the ice margin and can be seen clearly here.

10. Since the surface altitude $s(t, r)$ is known, there is no need for making a distinction between ice thickness and surface elevation (Eqs 30 and 31). These observations are exactly equivalent.

We disagree here. Since $b(r\_i)$ is not zero, there is a difference between these equations (see p.9) and, as the equations (31) and (32) now are defining the map between the model variables and the physical observations that are available, we feel that the distinction between these cases should be made.

11. Comment on the term 3-DVar.

We agree that the term '3DVar' is not exactly correct here. We have had difficulty in finding a short description of this method. However, the method does approximately minimize the nonlinear 3D-Var objective function under the assumption of (19) (or equivalently it minimizes the linearized version of the objective function), whilst keeping a fixed structure in the background error covariance. We note that in the moving mesh context, despite having a fixed structure, the background error covariances retain some flow dependence because the covariances depend on the positions of the mesh points, which move with the flow, as pointed out in section 4.2 of the paper. We have added a comment at the end of section 3.1.1 pointing out that the method we use is a variant of the traditional nonlinear 3D-Var algorithm, but that nevertheless we retain the term 3D-Var in the paper (p. 6, lines 25-27) : 'Although the scheme we propose here to use with the moving mesh method is a variant of the traditional nonlinear 3D-Var method, it is in essence a variational method with a fixed structure for the background covariance matrices and we will refer to it as the 3D-Var method in the rest of the paper.'

12. The right-hand side of Eq. (21) should read $\mathbf{B}k \, \mathbf{H}k^{\mathrm{T}}$ (…) (and not $\mathbf{B}k \, \mathbf{H}k$).

Corrected, with thanks (p.6 Eq.(21)).

13. P. 18, l.10. You mention $Tclim$. Reference should be made here to Eq. (A4).

It was already stated in Section 5.1 on p.19 line 10 and on p. 20 line 7 that the details of the mass balance equation are given in Appendix A2 and that the climate forcing is as defined in the Appendix A2. We have added the phrase (p. 20 line 1) : 'as defined in Eq (A4)' .

14. Eq. (A3). I think the unit for $T$ is kelvin. Say it.

The unit for T is $^{\mathrm{o}}$C , consistent with Table 2 in the Appendix A2.

**Response to Reviewer 2:**

General Comments —————————-
Bonan et al discuss a data assimilation (DA) technique applied to a moving mesh ice sheet model. The promise lies in its ability to add observations of ice sheet margin position, and indeed, correctly account for the motion of that margin. The paper employs a rather simplified 1D description of ice sheet physics, so I would tend to see it as sketch of a technique that might be useful in the more complex 2 or 3 D problems currently of interest. Given that, I would hope to be able to assess the value in developing such methods further, but the problem studied is just too far away from the problems of interest for me to feel any the wiser.

The research addresses the use of data assimilation with new numerical techniques for modelling moving boundary problems. We illustrate our approach on ice flow with the aim of efficiently obtaining more accurate estimates for the margins of ice sheets. A relatively simple model of ice flow is used here to investigate the new techniques. The new moving-mesh numerical methods for the ice flow have already been validated for both 1-D and 2-D models of ice flow (see [1] and [2]). The methods have also been applied to a number of other moving boundary problems, including tumour growth and chemical spreading, as referenced in the paper. The aim of this paper is to demonstrate that it is possible to combine sophisticated data assimilation methods with these moving mesh numerical modelling techniques. Given that the techniques are successful on the simplified problem, there is no reason that these cannot be extended to much more complex problems. The major advantage of the moving mesh method is that only a small number of mesh steps is needed to accurately determine the boundary positions of the flow, unlike adaptive and fixed grid mesh methods. We have reordered the introduction and added comments and additional references to emphasize the aims of the paper.

The ice sheet physics is the 'shallow ice approximation', which I think it is fair to say is of little interest in contemporary ice dynamics. It is still of (diminishing) interest in the study of the distant past (ie the rise and fall of ice ages), and it is possible to imagine this sort of technique being of great interest there *if* it could be used in the right kind of data assimilation. The data would be sparse in both space and time - isolated values ( _1 point in the whole domain) for past surface elevations of ice sheets where their surface intersected with rock, and some observations of their margins/extent through time from depositions, landscape scouring and so on. The synthetic data in this paper is very much more like contemporary satellite data – dense observations of surface velocity – only available at all through satellite observation – and elevation.

The data assimilation method combined with the moving mesh method works effectively even with sparse observations relative to the degrees of freedom, see [2][3] for examples, but the scenario with more dense satellite data that we have used here is more realistic today. For surface elevation observations the method remains accurate even for very few data. For observations of surface velocity, as expected, there is some degradation of the analysis because the derivatives in the observation operator are sensitive to the loss of data; nevertheless, the results remain satisfactory and informative. In any case, we note that measurements of velocity are associated with high quantities of satellite data, rather than with isolated core measurements or marginal observations.

It is possible that the shallow ice approximation (eq 4) is mathematically close enough to the systems of interest to imagine the DA methods being of wider value. I'm simply not sure. In 1D eq 4 would be replaced by a nonlinear elliptic equation in U(x) and in the most simple case a boundary condition on U(rl) from Schoof 2007 - which incidentally implies a non-zero flux across the margin (the grounding line) so that eq 7 for the mesh movement is not right. It is true that the elliptic equation can be approximated by something like eq 4 far from the margin, but that is invalid in the fast sliding glaciers that are seeing present day change. Does that fact that eq 4 should be an elliptic PDE, rather than a

simple expression, matter to the conclusions of the paper? It might. It means that eq 34 also requires the solution to an elliptic PDE, so that in turn it is harder to find dH/dx_h and dH/dx_r in eq 27. That is possible though, indeed, most recent progress in ice sheet DA involves such calculations. It also must have some impact on the ETKF method, because each member is much more expensive - how well will ETKF perform if the number of samples is limited? The 200 members used here might be the practical limit in a 2D or 3D problem, but there would be more degrees of freedom. I would have been interested to see how well/poorly ETKF performed with, say, 20 members

The fact that an elliptic problem needs to be solved at each step of the model is the same for any other model of the flow. Here we use a direct solution in the simplified case, but this is not necessary. The elliptic problems can be solved in parallel numerically at each time step in the ensemble filter assimilation method, but if computational power is a constraint, then the 3DVar method gives good results without solving multiple elliptic problems. Models incorporating flux across the margin can also be treated directly.

Regrettably, we have not been able to include further experiments using fewer ensembles, as originally intended. Our first author has had to withdraw from research due to serious family illnesses. In any case, it is known for the ETKF method that some form of localization as well as inflation is needed in the case where there are significantly fewer ensemble members than degrees of freedom and we feel that examining this issue is beyond the scope of this paper. However, we think that the revised paper now has sufficient material of interest for publication and so we are resubmitting the paper for consideration.

Overall, I'd say that unless the general mathematics of the moving mesh and DA approach are interesting in themselves (hard for me to judge, but they seem to be relatively straightforward), then this work needs to be based on more relevant ice sheet physics.

The moving mesh method has a wide range of applications other than ice-sheet dynamics and the ability to use data assimilation with these methods is important. Data assimilation with other methods for tracking moving boundaries is very expensive, so we feel that the ability to track fronts accurately and efficiently is a major achievement. The simplicity and elegance of our approach is a particular advantage of the technique.

[1]  Bonan et al, The Cryosphere, 2015.  http://www.the-cryosphere.net/10/1/2016/

[2]  Partridge, PhD Thesis, University of Reading, 2013.
http://www.reading.ac.uk/web/files/maths/DP_PhDThesis.pdf

[3]  Partridge D, Baines MJ, Nichols NK,  A Moving Mesh Approach to a Shallow Ice Glacier Model Incorporating Data Assimilation, University of Reading, Dept of Maths & Stats, Mathematical Report Series 1/2014.  
[revised manuscript text omitted]

---

## Referee Report (RR1)

Thanks to the clear explanations given by the authors, I think I now understand what they have exactly done. I consider the paper is now acceptable for publication. At the same time, I consider a number of modifications are still desirable, either in the scientific presentation or in the edition of the paper. Many of my comments and suggestions below could have been made on the previous version of the paper. I did not make them then either because they escaped my attention or because I considered they were of secondary importance at that stage.

1. As shown in particular by the fact that there is no increase of error in the forecasts that follow the assimilations (Figs 15, 16, 18), the system is extremely stable. Actually, the state of the system seems to be stationary over the time period (10-20 years) considered in Section 5 (the reference does not evolve in Figs 13 and 16). And the evolution of the reference is smooth (and presumably highly predictable) over the longer time periods (1000-2000 years) considered in Section 4. That makes the assimilation problem relatively easy. The authors stress that the originality of their paper is that it defines an approach for assimilation in a system with moving boundaries. That is true, and the paper is a significant contribution. But the problem would presumably much more difficult in a system where the evolution of boundaries depended on instabilities (as for instance in the motion in ice sheets with the possibility of sudden surges). I think it must be said that the problem studied in the paper is in a sense relatively easy. That does not degrade the value nor the significance of the paper.

2. Time scales considered in Section 4 and 5 differ by two orders of magnitude. It is not clear why (at least to me). Is it because the authors want to study in Section 5 the case of a rapid warming ? In any case, the similarity of the results obtained over so different time scales confirms the strong stability of the system.

3. Figure 3 shows the background error covariance matrices used in 3D-Var at analyses times $t_1$ = 500 yr and $t_2$ = 1500 yr. If the information is available, it would interesting to show how the actual background errors compare with the figure (just as, for ETKF, the spread of the ensemble is compared to the reference in Fig. 7 and other figures that follow). A similar remark applies to other figures showing estimated errors as produced by the assimilation, such as Figs 6 and 8.

4. Subsection 4.4. My understanding is that only observations of ice thickness are used there (at least, it is what the caption of Fig. 8 suggests). Say it clearly in the text from the start.

5. P. 24, ll. 2-3 (comparison between Figs 14 and 17), *We observe that the standard deviations for the node positions are smaller in the middle of the ice sheet than in the previous experiment.* Nothing of that sort is clearly visible from the figures.

6. P. 7, l. 2, *The Ensemble Kalman Filter* […] *aims to approximate the Extended Kalman Filter.* That is not really true. The Extended Kalman Filter is based on a local linearization of the evolution equation (16). The Ensemble Kalman Filter avoids any such linearization, and cannot be said to 'approximate' the Extended Kalman Filter.

7. P. 8, ll. 4-5, *Estimates obtained by* […] *using a moving-point numerical model provide more information* […] *than if we were using a fixed-grid method.* That is certainly true as concerns the position of the ice margin. But is it true of anything else ?

8. P. 6, l. 12. Say that the error in the background is assumed to be uncorrelated with the errors in the observations.

9. Eq. (19), $\mathcal{H}_k(x) \approx \mathcal{H}_k(x_k^b) + \mathbf{H}_k(\dots)$

10. P. 8, ll. 15 and 16. From what I understand, the words *state variables* are to be replaced by *ice thicknesses*. The same correction is to be made in other places (*e.g.*, p. 7, last line). Check carefully.

11. Eqs (31-34). The authors assign numerically defined 'default' values (0 or $b(r_i)$) to observations when they actually mean there are no observations at all. In the case of Eq. (31) for example, the observation reduces to the scalar value $h_i$ + … . (I do not think there is a real danger of confusion there, but it is preferable to be consistent).
The same correction may have to be made elsewhere. Please check.

12. Is it possible to give a reference for the rather complicated form of the discretization in Eqs (35-36) ?

13. The authors confirm that $T_0$ is expressed in °C in Eq. (A3). That is difficult to believe. It means that a slight variation in $T_0$ (which is said to be equal to -6 °C) would result in a large variation in Abl($t, r$).

14. $T_{clim}(t)$ in Eq. (A3) does not seem to be defined.

15. P. 22, l. 8, *since the background state is smaller than the reference state*. I rather suggest *since observations may lie outside of the spatial range covered the background estimates of the node positions.*
And, three lines below … , *since experience shows that there is always at least one member of the ensemble which lies outside of the range covered by the observations.*

16. P. 14, caption of Fig. 15, l. 2, … *in Fig. 4,* …

17. P. 4, l. 12, *One of the points is dedicated to the static ice divide r = 0 , while another point tracks* …. Do you mean *One of the integral upper bounds is the current point $r^\wedge(t)$, while the other is the margin position $r_l(t)$* ? But that remark seems to me of no real interest.

18. P. 18, l. 1, *This effect would not necessarily appear with another set of observations*. The same remark would probably be true of many other results in the paper. I suggest you remove that sentence.

19. Caption of Fig. 18, last line, … *3D-Var gives better results for the ice thickness at r = 0*. I think this is to be put in the text.

20. P. 24, l. 20, sentence starting *This can be achieved either by using an appropriate flow-dependent background covariance matrix* … My understanding is that this is empirical. I suggest *This can be empirically achieved* … Actually, both the maintenance of positive thicknesses and of an ordered mesh is obtained empirically. A similar correction may have to be made at other places in the paper.

---

## Editor Decision (ED1)

Dear Nancy,

Two referees have now sent their reports on the revised version of your paper. They are the same as the referees of the first version. In particular, referee 2, who has again let his name known, is Stephen Cornford.

Referee 2 considers your paper is acceptable as it is. Referee 1 also considers it is acceptable. But he makes a number of suggestions for modifications, which have to do with the presentation of the paper much more than with its scientific content.

Please revise your paper considering all of referee 1's suggestions. When you send the new version, please state how you have responded to all of these suggestions. I do not intend to submit your paper to further review.

I thank you for having submitted your paper to *Nonlinear Processes in Geophysics*, and I look forward to receiving your new version,

With regards,

Olivier Talagrand
Editor
*Nonlinear Processes in Geophysics*

---

## Author Response (AR2)

**Response to Comments of Reviewers:  npg-2016-45**

We thank the reviewers and the editor for their careful reading of the paper and the detailed comments we have received.  We are pleased that the discussion and the improvements to the paper have increased understanding of the research.

**Reviewer 1:**  We have made further corrections to the paper as suggested by Reviewer 1, as follows:

Thanks to the clear explanations given by the authors, I think I now understand what they have exactly done. I consider the paper is now acceptable for publication. At the same time, I consider a number of modifications are still desirable, either in the scientific presentation or in the edition of the paper. Many of my comments and suggestions below could have been made on the previous version of the paper. I did not make them then either because they escaped my attention or because I considered they were of secondary importance at that stage.

1. As shown in particular by the fact that there is no increase of error in the forecasts that follow the assimilations (Figs 15, 16, 18), the system is extremely stable. Actually, the state of the system seems to be stationary over the time period (10-20 years) considered in Section 5 (the reference does not evolve in Figs 13 and 16). And the evolution of the reference is smooth (and presumably highly predictable) over the longer time periods (1000-2000 years) considered in Section 4. That makes the assimilation problem relatively easy. The authors stress that the originality of their paper is that it defines an approach for assimilation in a system with moving boundaries. That is true, and the paper is a significant contribution. But the problem would presumably much more difficult in a system where the evolution of boundaries depended on instabilities (as for instance in the motion in ice sheets with the possibility of sudden surges). I think it must be said that the problem studied in the paper is in a sense relatively easy. That does not degrade the value nor the significance of the paper.

We have recognised and stated in the introduction that the model used is a relatively simple model of the ice sheet dynamics (see p 2 lines 11-12).  However, the model is a highly nonlinear diffusion equation that  might be expected to introduce problems in some circumstances.  The paper demonstrates that for the given conditions the assimilation method works well and that there is potential for other applications.

2. Time scales considered in Section 4 and 5 differ by two orders of magnitude. It is not clear why (at least to me). Is it because the authors want to study in Section 5 the case of a rapid warming ? In any case, the similarity of the results obtained over so different time scales confirms the strong stability of the system.

It is correct that we are looking at a rapidly warming climate, as described by the equations in Appendix A2  -  see also response to Comment 13.   The problem is realistic as the available observation records roughly span 20 years.  To make the objective clear we have added a sentence in the introduction to Section 5 (p.19 line 8) stating that  'We investigate the case of a rapidly warming climate over a short time-scale.'

3. Figure 3 shows the background error covariance matrices used in 3D-Var at analyses times $t1$ = 500 yr and $t2$ = 1500 yr. If the information is available, it would interesting to show how the actual background errors compare with the figure (just as, for ETKF, the spread of the ensemble is compared to the reference in Fig. 7 and other figures that follow). A similar remark applies to other figures showing estimated errors as produced by the assimilation, such as Figs 6 and 8.

Unfortunately this information is not directly available from the 3DVar process, which only produces a single estimate of the state of the system.  The error between the background and the reference and between the background and the analysis can be seen at each assimilation time from the figures 4, 5.

4. Subsection 4.4. My understanding is that only observations of ice thickness are used there (at least, it is what the caption of Fig. 8 suggests). Say it clearly in the text from the start.

That is correct - this has been stated in the introductory section 4.2 (see p 11, lines 9-10).   All the subsequent experiments use the same data, as stated in each section, with the exception of section 4.6 where we include an observation of the margin, as is explained.

5. P. 24, ll. 2-3 (comparison between Figs 14 and 17), *We observe that the standard deviations for the node positions are smaller in the middle of the ice sheet than in the previous experiment.* Nothing of that sort is clearly visible from the figures.

We agree and have removed this statement.

6. P. 7, l. 2, *The Ensemble Kalman Filter* […] *aims to approximate the Extended Kalman Filter.* That is not really true. The Extended Kalman Filter is based on a local linearization of the evolution equation (16). The Ensemble Kalman Filter avoids any such linearization, and cannot be said to 'approximate' the Extended Kalman Filter.

Agreed, but it should be recognized that the Ensemble Filter is only an approximation since the state estimate is found in only a subspace of the domain.  We have reworded this sentence to read (see p. 7 line 2):   'The ensemble Kalman Filter (EnKF) introduced by Evensen (1994) approximates a fully nonlinear Monte Carlo filter.'

7. P. 8, ll. 4-5, *Estimates obtained by* […] *using a moving-point numerical model provide more information* […] *than if we were using a fixed-grid method.* That is certainly true as concerns the position of the ice margin. But is it true of anything else ?

Yes, it gives more accurate estimates of the thickness of the ice at more accurately defined positions throughout the region.

8. P. 6, l. 12. Say that the error in the background is assumed to be uncorrelated with the errors in the observations.

Done  -  phrase added to p. 6 line 13 (before eq (19)).

9. Eq. (19), $H_k (x) \approx H_k (x_k^b) + \mathbf{H}_k (…)$

Corrected with thanks.

10. P. 8, ll. 15 and 16. From what I understand, the words *state variables* are to be replaced by *ice thicknesses*. The same correction is to be made in other places (*e.g.*, p. 7, last line). Check carefully.

Thank you for noting this. We have replaced the term 'state variables' by the term 'model variables' to distinguish these from the nodal state variables, see pages 7 (lines 24 & 28) and 8 (lines 15-16). The method applies to other models where the model states are not necessarily ice-thickness. (See section 5 for example)

11. Eqs (31-34). The authors assign numerically defined 'default' values (0 or $b(ri)$) to observations when they actually mean there are no observations at all. In the case of Eq. (31) for example, the observation reduces to the scalar value $hi + \ldots$ . (I do not think there is a real danger of confusion there, but it is preferable to be consistent). The same correction may have to be made elsewhere. Please check.

This point is obscure to us. The observation operator maps the state of the system to the observation. In this case the equations map the values of the state variables from the grid to an observation of the thickness at a point r^o. If r^o is not in the given interval then the observation is assumed to be at the base bedrock, which would be correct at the end of the domain.

12. Is it possible to give a reference for the rather complicated form of the discretization in Eqs (35-36) ?

This is a classic upwind 1-step discretization in space of the nonlinear equation (4) where $b$ is a function of $r$ and $n=3$. Further information is given in Appendix B2 of the paper by Bonan et al, The Cryosphere, 2016, where the moving mesh numerical method is described. We have cited this reference now on p. 9 line 24.

13. The authors confirm that $T0$ is expressed in °C in Eq. (A3). That is difficult to believe. It means that a slight variation in $T0$ (which is said to be equal to -6 °C) would result in a large variation in $Abl(t, r)$.

We expect this to be the case as here we are investigating the case of a rapidly warming climate over a short time-scale.

14. *Tclim(t)* in Eq. (A3) does not seem to be defined.

T_clim is different depending on the application and is defined in section 5.1 in words, first for the derivation of the initial state for the reference (p. 19 lines 15-16 & p. 20 lines 1-2) and then for the experiment (p. 20 lines 4-6) where T_clim = 6 + 0.2 t. We have introduced the equation for the latter in brackets to clarify the equation used in the experiments.

15. P. 22, l. 8, *since the background state is smaller than the reference state*. I rather suggest *since observations may lie outside of the spatial range covered the background estimates of the node positions.*
And, three lines below … , *since experience shows that there is always at least one member of the ensemble which lies outside of the range covered by the observations.*

The remarks here refer to the case being discussed and not to general points about the methods. We have added phrases to indicate this and to make clear that it is the domain of the background that is smaller than that of the reference

16. P. 14, caption of Fig. 15, l. 2, … *in Fig. 4, …*

Corrected, in caption of Figure 5, p.14.

17. P. 4, l. 12, *One of the points is dedicated to the static ice divide r = 0 , while another point tracks ….* Do you mean *One of the integral upper bounds is the current point r^(t), while the other is the margin position rl(t)* ? But that remark seems to me of no real interest.

The point is that the ice divide is static at r = 0 and the moving margin at r_l(t)  has the velocity given by the equation following.

18. P. 18, l. 1, *This effect would not necessarily appear with another set of observations*. The same remark would probably be true of many other results in the paper. I suggest you remove that sentence.

Agreed - the sentence has been removed.

19. Caption of Fig. 18, last line, … *3D-Var gives better results for the ice thickness at r = 0.* I think this is to be put in the text.

Agreed.  The sentence in the last paragraph of section 5.3 has been reworded : 'As in previous experiments the ETKF performs better than 3D-Var for the position of the margin, but 3D-Var gives better results for the ice thickness at r = 0 and performs reasonably well overall in this nonlinear context.'

20. P. 24, l. 20, sentence starting *This can be achieved either by using an appropriate flowdependent background covariance matrix …* My understanding is that this is empirical. I suggest *This can be empirically achieved …* Actually, both the maintenance of positive thicknesses and of an ordered mesh is obtained empirically. A similar correction may have to be made at other places in the paper.

Agreed.  The word 'empirically' has been added

[revised manuscript text omitted]